# RAT+: Train Dense, Infer Sparse - Recurrence Augmented Attention for Dilated Inference

Xiuying Wei [1]   Caglar Gulcehre [1]

## Abstract

Structured dilated attention has an appealing inference-time efficiency knob: it reduces the FLOPs of attention and the KV cache size by a factor of the dilation size $D$, while preserving long-range connectivity. While prior work studies it by training each configuration from scratch, directly sparsifying a pretrained attention model into a dilated pattern leads to severe accuracy degradation, preventing flexible reuse across inference scenarios. We introduce **RAT+**, a dense-pretraining architecture that augments attention with *full-sequence recurrence* and *active recurrence learning*. A single RAT+ model is pretrained densely once and can then be flexibly switched at inference time to dilated attention (optionally with local windows) or hybrid layer/head compositions, requiring only a short 1B-token resolution adaptation rather than retraining separate sparse models. At 1.5B parameters trained on 100B tokens, RAT+ closely matches dense accuracy at $D = 16$, and drops by about 2–3 points at $D = 64$ on commonsense reasoning and LongBench tasks. We further scale to 2.6B and 7.6B parameters and observe even more promising performance (e.g., a 1-point average accuracy loss with a $64\times$ reduction in attention FLOPs and KV cache size). Code is available at https://github.com/wimh966/rat-plus.

## 1. Introduction

Efficiency has become increasingly important for modern language models, as standard attention (Vaswani, 2017) incurs a quadratic cost in both FLOPs and memory with respect to sequence length. Among many orthogonal approaches to improving efficiency (e.g., low-bit representations (Jacob et al., 2018) or faster optimization (Jordan, 2023)), a major line of work focuses on sparsifying or replacing the attention mechanism, which can be broadly grouped into two categories: training-from-scratch sparse architectures and inference-time sparsification. In this work, we instead explore training a more capable dense model to enable more flexible sparse inference patterns.

Regarding efficient architectures trained from scratch, structured dilated attention (Ding et al., 2023; Beltagy et al., 2020; Zaheer et al., 2020) with carefully designed patterns was previously very popular due to its high efficiency and direct long-range access. It has been revisited in recent years, often coupled with other modules, such as in DeepSeek's NSA (Yuan et al., 2025) and the RAT model (Wei et al., 2025). RAT chunks the input sequence, applies a forget-gate-like recurrence within chunks, and then applies dilated attention across chunks, achieving performance on par with dense models at a dilation size of 16. Other efficient solutions include state space models (Gu & Dao, 2023; Dao & Gu, 2024), linear attention (Katharopoulos et al., 2020; Yang et al., 2024), and grouped-query attention (Ainslie et al., 2023; Shazeer, 2019). Despite the strong performance of these approaches, a major limitation is that each design configuration (e.g., dilation size, number of KV heads, or state size) typically requires separate training, and thus can be costly and inflexible in satisfying different downstream efficiency–accuracy trade-offs.

Another line of work adopts inference-time sparsification on pretrained attention models, including local window patterns (Xiao et al., 2023) and importance-based methods such as top-k block attention (Tang et al., 2024; Lu et al., 2025). However, sparsifying into a dilated pattern suffers from severe performance degradation, despite being a particularly attractive design. It reduces prefilling FLOPs (which methods such as (Li et al., 2024; Zhang et al., 2023) do not), decoding FLOPs, and KV cache storage (which top-k block attention does not reduce), while preserving long-range access (which local window attention fails to maintain).

Comparing the success of training dilated attention from scratch with its failure at inference-time sparsification, we are motivated to address this gap by designing a more capa-

[1]CLAIRE lab at EPFL, Lausanne, Switzerland. Correspondence to: Xiuying Wei <xiuying.wei@epfl.ch>, Caglar Gulcehre <caglar.gulcehre@epfl.ch>.

ble dense model that augments attention with lightweight recurrence. RAT+ can flexibly adapt to different levels of attention sparsity in terms of dilation at inference time. Technically, our analyses first reveal the need for an explicit mechanism, such as recurrence, to construct a complete receptive field for dilated attention. To enable effective inference-time sparsification, we further propose using overlapped chunk size that simplifies to full-sequence recurrence, along with active recurrence learning to ensure consistent recurrence behavior and sufficient optimization.

In experiments, we demonstrate that a pretrained RAT+ model can be adapted to different dilated attention configurations (with optional local attention), as well as hybrid variants across layers and heads, using only 1B tokens for resolution adaptation. These variants maintain strong performance on short-context commonsense reasoning tasks (Gao et al., 2024) and LongBench (Bai et al., 2023). For example, at the 7B model scale, $\mathtt{D} = 64$ incurs only a 1-point accuracy drop while achieving over $40\times$ higher full-model maximum throughput when decoding 1K tokens with a context window of 16K. Moreover, we show that the recurrence in RAT+ can improve top-k block attention compared to standard attention on needle-in-a-haystack (NIAH) tasks in the RULER (Hsieh et al., 2024) benchmark.

Our main contributions include:

1. Compared to existing efficient architecture designs, we propose a new paradigm for exploring efficiency by designing a more capable dense model that enables more flexible inference-time behavior. RAT+ supports various dilated attention and their hybrid variants (with only a short 1B-token resolution adaptation), and also improves top-k block attention performance.

2. In RAT+, we propose two novel techniques: full-sequence recurrence and active recurrence learning to enable flexible inference-time sparsification for dilated attention. Our design is grounded in detailed analyses, revealing both the motivation for recurrence and the key challenges for dilated inference.

3. We show that dilated attention is an effective inference-time sparsity pattern, achieving strong efficiency, accuracy, and scalability. RAT+ maintains strong results across short-context, long-context, and retrieval-heavy tasks. We further observe that scaling to larger sizes (e.g., a 2.6B model trained on 200B tokens and a 7.6B model) narrows the performance gap between dense and dilated attention, demonstrating the scalability of RAT+.

## 2. Preliminaries

We denote the sequence length by $\mathtt{T}$, dilation size by $\mathtt{D}$, local window size by $\mathtt{W}$, chunk size by $\mathtt{L}$, and the number of blocks selected in the attention of the top-k block by $\mathtt{K}$. $\mathtt{D}^{\dagger}$ is used

to highlight the attention model without recurrence.

We distinguish the size of the chunk from the dilation size below, noting that the original RAT sets $\mathtt{L} = \mathtt{D}$. Within each chunk, RAT first applies a simple linear recurrence over the attention keys $\boldsymbol{k}$ and values $\boldsymbol{v}$, which reduces to forget-gate behavior with an input-dependent gate $\boldsymbol{g}$. For the token at position $l$,

$$\begin{aligned} \tilde{\boldsymbol{v}}_l &= \boldsymbol{g}_l \odot \tilde{\boldsymbol{v}}_{l-1} + (1 - \boldsymbol{g}_l) \odot \boldsymbol{v}_l, \\ \tilde{\boldsymbol{k}}_l &= \boldsymbol{g}_l \odot \tilde{\boldsymbol{k}}_{l-1} + (1 - \boldsymbol{g}_l) \odot \boldsymbol{k}_l. \end{aligned} \quad (1)$$

It has $\mathcal{O}(1)$ FLOPs and KV cache, and can be implemented through a one-step update in decoding and parallel scan in prefilling and training. Then, for a token at position $d$ in dilation block $b$, dilated attention is applied over the gated keys and values:

$$\boldsymbol{y}_{b,d} = f([\boldsymbol{q}_{b,d}\tilde{\boldsymbol{K}}_{:,-1}^{\top}; \boldsymbol{q}_{b,d}\tilde{\boldsymbol{k}}_{b,d}^{\top}])[\tilde{\boldsymbol{V}}_{:,-1}; \tilde{\boldsymbol{v}}_{b,d}], \quad (2)$$

corresponding to the pattern shown in Fig. 1(c), which reduces both FLOPs and the KV cache size by a factor of $\mathtt{D}$. We do not consider the sliding variant, as it reduces parallelism during prefilling and training, and has to keep the full KV cache for decoding. Also, note that RAT with $\mathtt{L} = \mathtt{D}$ and sparse training also fails in flexible dilated inference.

## 3. Motivation for recurrence

We analyze how the simple recurrence Eq. (1) enables effective dilated attention by bridging disconnected attention patterns. We study this from both the training-from-scratch and the inference-time sparsification perspectives.

**Training-from-scratch** We find that the recurrence used in RAT plays a critical role in its subsequent inter-chunk attention (which corresponds to dilated attention), and we attribute this to its ability to help provide a complete layer-wise receptive field.

To demonstrate this, we conducted experiments in Table 1 comparing dilated attention with and without recurrence, as well as with optional local attention. We observe that dilated attention without recurrence fails to properly optimize and finally converges to suboptimal plateaus during training, as expected given the disconnected attention graph. While adding local attention links neighboring tokens to form a conceptually connected attention graph, it must also serve a distinct primary role: selectively aggregating fine-grained information within the local window for the current token. We hypothesize that this dual responsibility limits the ability of local attention to reliably support effective graph connectivity in dilated attention. In contrast, recurrence explicitly and independently provides such support. Moreover, even dense attention benefits from the introduction of recurrence, as shown in the last block of Table 1.

*Table 1.* Results of training 1B-parameter models on 100B tokens from scratch, with or without the simple recurrence. All models are equipped with output gating (denoted as ogate below), (Hua et al., 2022; Qiu et al., 2025). Perplexity (PPL) is measured on a held-out 0.5B-token validation set with a 4K sequence length, and Avg. CSR denotes the average result across six commonsense reasoning tasks. Note that local attention typically achieves strong performance on short-context tasks while 7.96 at $D^\dagger = 64, W = 64$ is a high PPL. See Table 14 for implementation details and full results.

| Model | Val PPL T = 4096 | Avg. CSR T ≤ 300 |
|---|---|---|
| dilated attention ($D^\dagger = 8$) | 43.89 | Fail |
| RAT (L = 8, D = 8) | **7.52** | **57.54** |
| dilated attention ($D^\dagger = 16$) | 63.94 | Fail |
| RAT (L = 16, D = 16) | **7.60** | **56.99** |
| dilated attention ($D^\dagger = 64, W = 64$) | 7.96 | 57.40 |
| RAT (L = 64, D = 64, W = 64) | **7.63** | **58.74** |
| attention | 7.44 | 58.33 |
| attention + recurrence | **7.38** | **58.84** |
| attention - ogate | 7.61 | 57.54 |
| attention - ogate + recurrence | **7.52** | **58.23** |

**Inference-time sparsification** Although dense attention models can be sparsified into local attention or importance-driven patterns such as top-k block attention, they do not adapt effectively to dilated attention, even when equipped with local attention, as shown in Table 2. We further apply a light degree of fine-tuning to assess whether this issue can be mitigated.

First, lightly fine-tuning a sparsified dilated-attention pattern from a pretrained dense model yields a rapid loss decrease with only a small number of tokens; however, the gains remain similar when using more tokens. We view this initial rapid improvement as a *resolution-adaptation* process when switching from continuous dense attention to dilated connections. This effect is conceptually related to the fine-tuning used when adjusting positional encoding frequencies (Peng et al., 2023b; bloc97, 2023), but is simpler here, as it does not require attention length generalization. Second, because local attention in this setting is treated solely as an inference-time configuration and is not explicitly trained to construct a complete receptive field, there remains a clear performance gap relative to the training-from-scratch regime, potentially requiring prolonged, continued pretraining to close.

In general, results from both aspects highlight the importance of explicitly constructing a complete receptive field for dilated attention, for example, via simple recurrence Eq. (1). Although the efficient RAT architecture can address this problem in the pretraining setting, it does not extend directly to the inference-time sparsification regime. Next, we introduce an architecture designed to address this challenge.

*Table 2.* We tune the pretrained attention model with different numbers of tokens into the sparse variant (four initial tokens have already been added for attention sink (Xiao et al., 2023)) and refer to the initial phase with rapid loss reduction (within a few hundred million tokens) as the resolution adaptation process. After this phase, improvements exist but much slower, as other modules such as local attention or the FFN are required to mitigate the performance gap, often requiring substantially longer retraining to match training-from-scratch (e.g., $D^\dagger = 64, W = 64$ as 7.96 and $D^\dagger = 1$ as 7.44) performance. Since this resolution adaptation phase stems from the attention mechanism, it also transfers to RAT+. However, RAT+ does not require the later stage of prolonged training.

| Val PPL | 0 | 500M | 1B | 2B | 3B |
|---|---|---|---|---|---|
| $D^\dagger = 16, W = 16$ | 11.51 | **9.54** | 9.37 | 9.22 | 9.11 |
| $D^\dagger = 64, W = 64$ | 9.20 | **8.63** | 8.57 | 8.51 | 8.47 |
| $D^\dagger = 4$ | 4590 | **43.85** | 40.71 | 37.99 | 36.52 |

## 4. RAT+: Dense model for sparse inference

We now propose a simple and novel architecture based on RAT for inference-time sparsification, where a single dense model is trained once (e.g. D = 1) and effectively adapted at inference time to different sparse attention patterns, including dilated attention (e.g. D = 1, 2, 4, 8, 16, 32, 64).

The RAT architecture is insufficient for this purpose, as it is a sparse design originally developed for training from scratch and obviously lacks the dense attention capability required in D = 1. More concretely, the pretraining-free setting requires: (1) dense attention capacity together with a recurrence capacity that can reliably handle sequences of at least length 64 required in D = 64; and (2) recurrence outputs follow the consistent distributions across different configurations, so that subsequent dilated attention does not have to cope with mismatched input distributions, which would otherwise hinder flexible inference.

**Overlapped chunk size** The RAT architecture sets the chunk size equal to the dilation size. For example, a recurrence trained with length L = 64 must be adapted to L = 4 when performing inference at D = 4. While it can handle shorter sequences, we observe that recurrence outputs at early time steps exhibit substantially different distributions (Fig. 9), making subsequent dilated attention difficult to adapt to this distribution shift. Such discrepancies at initial steps have also been identified as a form of covariate shift in prior work (Cooijmans et al., 2016). Consistently, the ablation results in Table 12 show that adapting to small chunk sizes (i.e., initial time steps) leads to noticeably higher perplexity because of violating requirement (2).

Instead of applying normalization across time steps, which increases complexity and slows training, we use overlapped recurrence chunks with a fixed-length L for each token (i.e., each token is associated with a fixed-length recurrence window) throughout training and inference. During evalua-

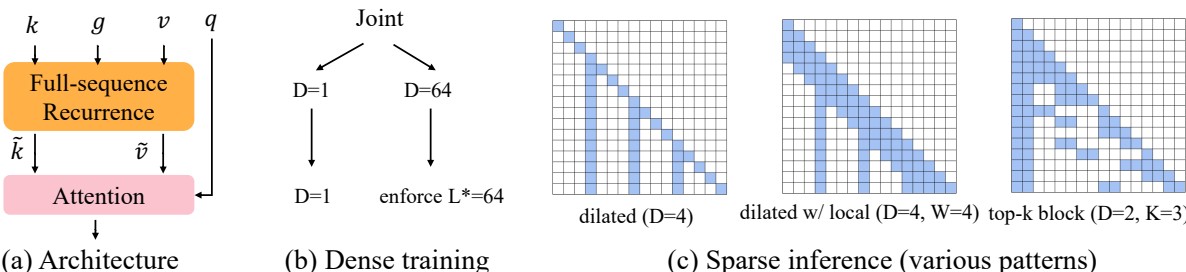

*Figure 1.* (a) For architectural simplicity, we adopt an extreme overlapped setting, i.e., full-sequence recurrence with $L = T$. (b) Joint training to preserve dense attention capability while enforcing active recurrence learning with desired effective length $L^* = 64$. (c) After pretraining, the resulting model can be efficiently adapted to various sparse inference patterns including effective results on dilated attention (optional local attention). It also has better performance on top-k block attention compared to attention.

tion, the dilation size is varied while keeping L fixed (e.g., L = 64), satisfying the requirement (2) with the consistent recurrence outputs across different dilation settings.

**Active recurrence learning**  We also need to satisfy requirement (1) by learning both dense attention and recurrence abilities. However, in the current architecture, the overlapped recurrence is hierarchically applied to the attention key and value activations before the scaled dot-product operation. We find that directly training it leads to lazy recurrence learning. Because full-sequence attention already provides complete connectivity, the model can rely on it regardless of the effective range learned by the recurrence. As a result, the recurrence has little incentive to pursue the intended length capability and instead converges to a shorter length that is easier to learn. Empirically, we observe in Table 12 that such models still exhibit high perplexity under sparse inference, though much better than attention.

To guarantee that the overlapped recurrence learns the required length (e.g., 64), we are motivated by the insight in Sec. 3 that a complete receptive field is critical for strong performance. We therefore introduce a stricter training configuration in which acquiring the desired recurrence capability provides a clear advantage over the shorter one. Concretely, we consider the sparse setting $L = 64, D = 64$, where insufficient recurrence capacity directly leads to degraded performance. Based on this, we adopt a simple joint training strategy that updates the model with a sparse case ($L = 64, D = 64$) and a dense case ($L = 64, D = 1$) per batch of data. The sparse case enforces active recurrence learning (ARL), while the dense case preserves sufficient attention capacity, ensuring strong performance in both dense and dilated settings. See Table 12 for a comparison with another training strategy.

**Simplification**  Moreover, active recurrence learning allows us to further simplify the architecture by extending the overlapped chunk size 64 to the sequence length without

performance loss, resulting in a full-sequence recurrence in implementation. This change reduces training to a single parallel forward scan over the sequence and simplifies KV cache management for recurrence. The minimum required recurrence capability (e.g., 64) is then enforced by sparse dilated attention during joint training, while longer-length ability is learned only when favored by the training loss. Consequently, we denote the full-sequence recurrence by $L = T$ and use $L^*$ for the actively enforced recurrence length when they differ. Fig. 7 and Subsec. 5.5 justify the choice of 64 based on the trade-off between the recurrence capacity and the flexibility of the dilation settings.

To conclude, with full-sequence recurrence and active learning, RAT+ achieves dense performance comparable to attention while remaining effective in different dilation settings (Table 3), with even stronger sparse results than training RAT directly, benefiting from dense FLOPs training.

*Table 3.* PPL performance of adapting different models to different dilation sizes with 1B tokens. Entries in the Train column marked with two settings denote training on both. For attention, we also include two variants trained under the joint training setting. The failures of RAT at small dilation sizes come from its lack of dense-attention capability and the use of chunk sizes equal to the dilation size. The gray row refers to the final RAT+ with full-sequence recurrence ($L = T$) and an active recurrence length of $L^* = 64$.

| Model | Train | Inference (D value) | | | | | | |
|---|---|---|---|---|---|---|---|---|
| | | 1 | 2 | 4 | 8 | 16 | 32 | 64 |
| **attention** | | | | | | | | |
| - | $D^\dagger = 1$ | 7.44 | 17.2 | 41.0 | 65.1 | 87.9 | 110 | 129 |
| - | $D^\dagger = 1, W = 64$ | 7.41 | 16.7 | 40.2 | 64.3 | 87.2 | 109 | 129 |
| - | $D^\dagger = 1, D^\dagger = 64$ | 7.82 | 16.2 | 34.7 | 53.0 | 70.9 | 90 | 110 |
| **RAT** (L = 16) | $D = 16$ | 8.41 | 8.08 | 7.79 | 7.64 | 7.61 | 7.74 | 7.93 |
| **RAT+** | | | | | | | | |
| $L = 64$ | $D = 1, D = 64$ | 7.4 | 7.42 | 7.45 | 7.48 | 7.52 | 7.58 | 7.67 |
| $L = T, L^* = 64$ | $D = 1, D = 64$ | 7.4 | 7.42 | 7.45 | 7.48 | 7.52 | 7.58 | 7.66 |

# 5. Experiments

In this section, we first demonstrate the effective dilated inference of RAT+ and position it among many existing

efficiency methods. We then report its scalability up to the 7B model scale and ablation studies.

## 5.1. Settings

Main results are based on 1.5B-parameter models trained on 100B tokens from FineWeb-Edu (Penedo et al., 2024), with a context length of 4096. In Subsec. 5.4, we further train 2.6B (100B and 200B tokens) and 7B models. All models adopt output gating with a corresponding linear projection (Hua et al., 2022; Qiu et al., 2025); RAT+ further includes an additional projection for recurrence. We leave parameter reduction for future work. *Unless otherwise specified (e.g., hybrid models), each sparse pattern is applied uniformly across all layers and used for both prefilling and decoding.*

Basic experiments include commonsense reasoning tasks (Gao et al., 2024), LongBench (Bai et al., 2023), and NIAH tasks from the RULER benchmark (Hsieh et al., 2024). Since NIAH consists of synthetic tasks with heavy prompts, we apply a supervised fine-tuning (SFT) stage by synthesizing training data, so that evaluation better reflects the models' intrinsic capabilities rather than prompt-following ability, which pretrained-only models are not optimized for. See Subsec. A.1 for full implementation details.

## 5.2. Main results

Accuracy and efficiency results are presented in this section, with NIAH results deferred to Subsec. B.3 and Table 16. *Here, we primarily compare with prior methods at similar efficiency levels which also reduce prefilling, decoding FLOPs and KV cache.*

*Table 4.* Performance of RAT+ on commonsense reasoning tasks. Models in the first two blocks are obtained by adapting pretrained dense models with a lightweight resolution adaptation of 1B tokens, while models in the third block are trained from scratch. Task definitions and evaluation metrics follow (Yang et al., 2024), and the results for the last two models are taken from (Dao & Gu, 2024; Yang et al., 2024). We report temporal-mixing FLOPs per token for each configuration. All tasks involve short contexts with $T \leq 300$; in particular, LMB. and Wino. have even shorter contexts. Thus large dilation settings almost reduce to a single recurrence. The $D = 128$ is supported by length generalization of the recurrence.

| Model | FLOPs $T \leq 300$ | ARC-C acc_n | ARC-E acc | Hella. acc_n | LMB. acc | PIQA acc | Wino. acc | Avg. - |
|---|---|---|---|---|---|---|---|---|
| **RAT+** ($L^* = 64$) | | | | | | | | |
| $D = 1$ | $T$ | 40.78 | 73.23 | 59.81 | 50.34 | 73.56 | 57.54 | 59.21 |
| $D = 2$ | $T/2$ | 41.47 | 73.23 | 59.55 | 50.36 | 73.34 | 56.51 | 59.08 |
| $D = 4$ | $T/4$ | 41.21 | 72.98 | 59.46 | 49.18 | 73.39 | 56.75 | 58.83 |
| $D = 8$ | $T/8$ | 40.96 | 72.98 | 59.09 | 48.52 | 73.23 | 54.70 | 58.25 |
| $D = 16$ | $T/16$ | 40.44 | 73.06 | 58.89 | 47.72 | 72.85 | 55.41 | 58.06 |
| $D = 32$ | $T/32$ | 40.53 | 73.19 | 58.20 | 45.74 | 72.85 | 56.83 | 57.89 |
| $D = 64$ | $T/64$ | 39.76 | 72.90 | 57.93 | 44.05 | 72.91 | 57.22 | 57.46 |
| $D = 128$ | $T/128$ | 39.59 | 73.19 | 57.37 | 41.8 | 72.91 | 57.69 | 57.09 |
| **attention** | | | | | | | | |
| $D^\dagger = 1$ | $T$ | 40.1 | 71.84 | 58.5 | 49.95 | 72.42 | 57.14 | 58.33 |
| $D^\dagger = 16$ | $T/16$ | 23.04 | 35.23 | 28.58 | 0.99 | 56.53 | 51.22 | 32.60 |
| **RAT** | $T/16$ | 39.76 | 72.6 | 56.95 | 46.03 | 72.14 | 54.46 | 56.99 |
| **Mamba2** | 256 | 37.88 | 72.47 | 55.67 | 45.66 | 71.87 | 55.24 | 56.47 |
| **GatedDeltaNet** | 288 | 38.39 | 71.21 | 55.76 | 46.65 | 72.25 | 57.45 | 56.95 |

**Does RAT+ preserve dense performance and enable stable dilation?** As shown in Table 3, RAT+ preserves perplexity for the dense pattern compared to attention-only models trained under the similar FLOPs budget. It also remains stable dilation, whereas attention-only models exhibit extremely high perplexity when directly sparsified into dilated attention (e.g., over 100 at $D = 64$). Downstream commonsense reasoning benchmarks for general language understanding, shown in Table 4, further show the feasibility of dilated inference. $D = 16$ achieves nearly the same average accuracy as dense attention ($D = 1$), and $D = 64$ incurs only about a 2-point drop while reducing temporal-mixing FLOPs by $64\times$.

*Table 5.* LongBench performance of different sparse inference patterns. Due to space constraints, other less informative tasks are deferred to Appendix Table 15 (e.g., similar performance on LCC or uniformly low scores on MusiQue). Since LongBench is prompt-heavy where pretrained-only models can be sensitive to, we focus on overall trends. Moreover, we present representative hybrid patterns to demonstrate feasibility rather than optimality, as optimal configurations depend on pretrained model properties. NQA: NarrativeQA; MF: MultiField-en; HQA: HotpotQA; 2WQA: 2WikiMultihopQA; RBP: RepoBench-P.

| **RAT+** ($T = 4096, L^* = 64$) | NQA | Qasper | MF | HQA | 2WQA | RBP | Avg. |
|---|---|---|---|---|---|---|---|
| **Dilated** (opt. local) | | | | | | | |
| $D = 1$ | 13.87 | **15.59** | **27.45** | 15.38 | 17.53 | 26.42 | 19.37 |
| $D = 2$ | **14.23** | **15.92** | **25.86** | **16.13** | **16.92** | 24.83 | 18.98 |
| $D = 4$ | 13.89 | 14.65 | 24.24 | 14.82 | 16.55 | 24.74 | 18.15 |
| $D = 8$ | **14.13** | 14.37 | **26.06** | 14.7 | 16.26 | 24.37 | 18.32 |
| $D = 16$ | **14.72** | 14.54 | **25.99** | 14.68 | **17.4** | 23.68 | 18.50 |
| $D = 32$ | 12.9 | 13.81 | 25.16 | 13.38 | **17.24** | 21.98 | 17.41 |
| $D = 64$ | 13.89 | 13.99 | 21.57 | 13.18 | 14.7 | 22.03 | 16.56 |
| $D = 128$ | 12.76 | 12.35 | 21.32 | 12.02 | 16.96 | 22.65 | 16.34 |
| $D = 16, W = 256$ | **14.06** | 13.69 | 24.75 | 14.15 | **17.9** | **27.49** | 18.67 |
| $D = 8, W = 512$ | **14.3** | **15.23** | 24.44 | **15.79** | 17.57 | 26.57 | 18.98 |
| $W = 1024$ (StreamingLLM) | 11.7 | 14.12 | 21.68 | 14.68 | 13.77 | 26.09 | 17.01 |
| $W = 256$ (StreamingLLM) | 11.34 | 12.18 | 19.38 | 12.84 | 15.55 | **26.22** | 16.25 |
| **Hybrid** (layer-wise) | | | | | | | |
| $D = 16$ (even layers) $\mid W = 1024$ | 12.97 | 14 | 22.01 | 13.63 | **17.51** | 26.01 | 17.69 |
| $D = 1$ (0, 12) $\mid D = 16, W = 256$ | 13.46 | 14.3 | **26.53** | **15.08** | 16.36 | 26.39 | 18.69 |
| **Hybrid** (head-wise) | | | | | | | |
| $D = 1$ (first two) $\mid D = 4$ | **14.54** | **15.59** | 25.56 | **17.31** | **18.86** | 23.73 | 19.27 |
| **Baseline to** $D = 16$ | | | | | | | |
| Mamba2 | 11.1 | 11.3 | 18.6 | 11.8 | 15.1 | 20.6 | 14.75 |
| GatedDeltaNet | 14.1 | 14.0 | 23.3 | 13.7 | 14.4 | 22.1 | 16.94 |
| **NTK extended** ($T = 16384$) | | | | | | | |
| $D = 1$ | 14.74 | 18.5 | 31.27 | 16.83 | 16.52 | 25.25 | 20.52 |
| $D = 4$ | 14.34 | 17.39 | 30.96 | 15.99 | 17.9 | 24.58 | 20.19 |

**Does RAT+ support various sparse inference patterns?** We then evaluate on LongBench, whose long-context setting enables a more meaningful study of different dilation patterns. A key finding is that different downstream tasks favor different patterns, supporting our train-dense, infer-sparse paradigm. For example, StreamingLLM underperforms ours on all tasks except RBP, suggesting a preference for local window patterns in that case. Incorporating local windows within or across layers can recover the performance for dilated attention. Inserting a small number of dense layers further benefits tasks such as MF.

We emphasize that our goal for hybrid sparse inference patterns is to demonstrate feasibility rather than optimality.

For instance, D = 16 interleaved with W = 1024 achieves intermediate performance, improving over local-window-only models (e.g., +4 on 2WQA), but does not consistently outperform dilated attention. This is because inference-time hybrid configurations are constrained by pretrained specialization, where different layers and heads vary in their suitability for sparsification. The search for optimal hybrid designs is left to future work. We further present results on length generalization using NTK-based RoPE scaling (bloc97, 2023), which extends the sequence length with either proportional FLOPs increase or intact FLOPs via dilation.

**What are the actual latency/throughput wins?** While linear projections dominate latency in short-context prefilling, the temporal-mixing operator becomes the main bottleneck during decoding—even at moderate sequence lengths (e.g., 4K)—as well as in long-context prefilling, where latency is inherently higher and thus more critical Fig. 2. The RAT+ operator with D = 16 achieves up to $6.3\times$ and $8.5\times$ speedups for hidden dimensions 2048 and 4096, respectively, at a prefilling length of 262K. When benchmarking the full temporal-mixing block (including linear layers), the measured speedup is reduced due to the additional recurrence projection and existing linear projections. Nevertheless, RAT+ still achieves $5.5\times$ and $6\times$ improvements over the attention block, as shown in Table 22 and Table 24. The overhead of RAT+ in prefilling arises from full-sequence recurrence implemented via *associative scan*, the use of *FlexAttention*, and the additional linear layer. Further optimization of these components may improve efficiency.

Compared to prefilling, the efficiency benefits during decoding are larger (see Table 23 and Table 25), as it only requires a simple one-step recurrence, adopts the same *FlashAttention* with shorter sequence lengths, and the extra linear projection contributes relatively little to latency. We observe that RAT+ can achieve wall-clock speedups close to the dilation factor, with around $13\times$ and $14.0\times$ at both the operator and block levels for dilation 16, especially when GPU resources are sufficiently utilized.

Finally, KV cache storage is another critical efficiency factor, as it directly limits the maximum batch size during decoding, which is typically memory-bound. Increasing the batch size is therefore key to improving GPU utilization. We further benchmark end-to-end maximum throughput on full models with 1.5B and 7.6B parameters. As shown in Fig. 4, D = 64 achieves over $60\times$ and $40\times$ higher throughput, respectively.

**Does recurrence help beyond dilation (top-k block)?** Beyond dilated attention, we find that RAT+ also benefits top-k block attention. Studies in Fig. 3, Table 17 and Table 18 show that using top-k block pattern in the dense RAT+ largely surpasses the performance when using in standard attention. The ablations show that the gain comes from recurrence. We hypothesize that improved block scoring is one key factor and provide head-level analysis in Table 9. For example, in Fig. 3, all dense configurations achieve perfect accuracy, whereas sparse variants from RAT+ maintain much higher accuracy than attention (e.g., 93.8 vs. 63.2 on NIAH-MK-2 at D = 64, K = 16). Moreover, ablating active recurrence learning during SFT which captures task-specific information leads to intermediate performance, indicating that the gains indeed come from recurrence. Consistent findings across other NIAH tasks and an additional top-k block attention method are reported in Subsec. B.3.

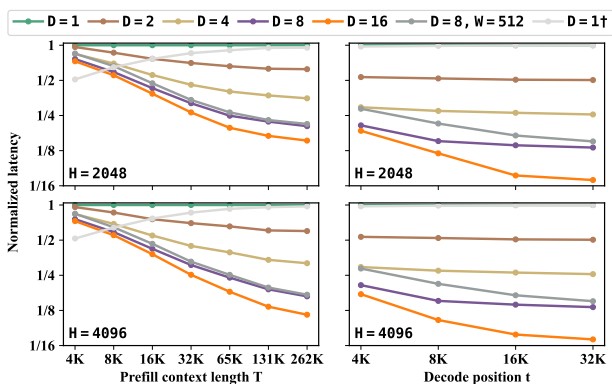

*Figure 2.* Efficiency results of the temporal-mixing operator on a single GH200 GPU, covering both prefilling and decoding scenarios with hidden dimension H. Prefilling latency is measured on sequences of 262K tokens. Decoding latency is measured for 256 or 128 batches of tokens for the two hidden dimensions, respectively; the baseline runs out of memory beyond 32K tokens. We use *FlexAttention* (Dong et al., 2024) for prefilling (with D = 1 reducing to FlashAttention), and *FlashAttention* (Dao et al., 2022)for decoding. Recurrence is implemented by *associative scan* in PyTorch for prefilling and a simple step update for decoding. Additional results are reported in Subsec. B.4.

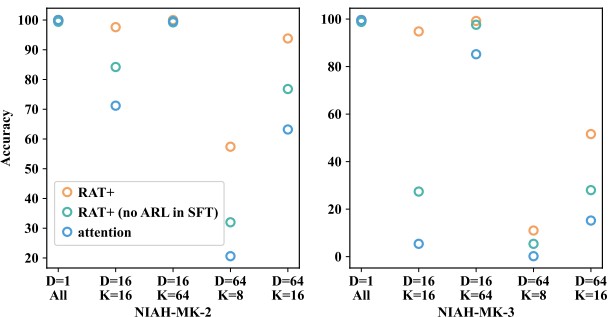

*Figure 3.* Results of top-k block attention with block size D and number of selected blocks K on the hard NIAH-MK-2 and NIAH-MK-3 tasks from the RULER benchmark with T = 4096. RAT+ (no ARL in SFT) means disabling active recurrence learning during the SFT, which further demonstrates the benefit of the recurrence. More results include remaining tasks in Table 17, MoBA-style top-k block attention in Table 18.

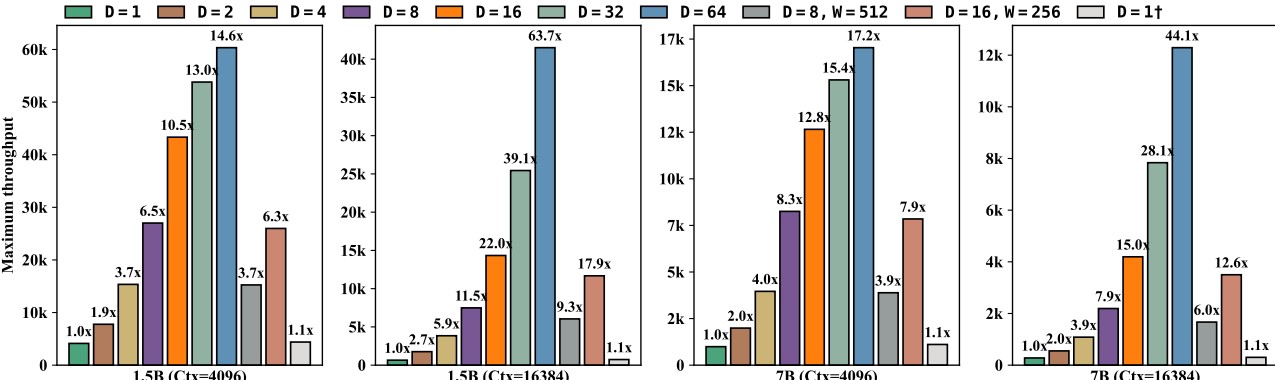

*Figure 4.* Maximum decoding throughput (tokens/sec) of the full 1.5B and 7B models for decoding 1024 tokens, measured at context lengths of 4096 and 16384, corresponding to prefilling lengths of 3072 and 15360 tokens, respectively.

*Table 6.* Position of RAT+ among existing efficiency methods, considering aspects including temporal-mixing prefilling FLOPs reduction, decoding FLOPs reduction, KV cache reduction, and direct long-range access. We define direct long-range access as the ability to explicitly attend to distant tokens. SSM and StreamingLLM, which operate at similar efficiency levels, are compared in Subsec. 5.2. Further comparisons are provided in Subsec. 5.3. Top-k block methods include Quest and MoBA.

| Method | Prefill | Decode | KV cache | Long-range access |
|---|---|---|---|---|
| RAT+ | ✓ | ✓ | ✓ | ✓ |
| SSM | ✓ | ✓ | ✓ | ✗ |
| GQA | ✗ | ✗ | ✓ | ✓ |
| StreamingLLM | ✓ | ✓ | ✓ | ✗ |
| Top-k block | ✓ | ✓ | ✗ | ✓ |
| SnapKV | ✗ | ✓ | ✓ | ✓ |

## 5.3. Positioning RAT+

In Subsec. 5.2, we compare with SSM and local window attention (StreamingLLM), which operate at similar efficiency levels. We further provide results for other training-from-scratch and inference-time sparsity approaches to better position RAT+ among existing efficiency methods Table 6.

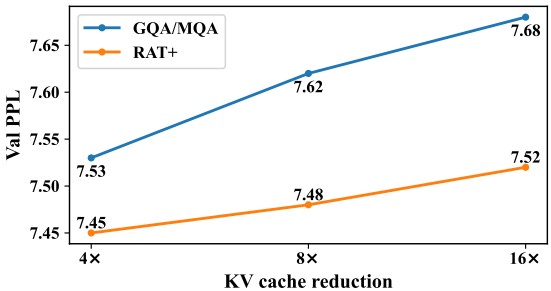

*Figure 5.* Comparison with GQA/MQA using different numbers of KV heads. Joint training is also applied to GQA/MQA ($D^{\dagger} = 1$, $W = 64$) to match training FLOPs. RAT+ achieves lower PPL and is more flexible: it requires only a single pretraining run and allows the local KV cache size to be configured at inference time.

**Comparison with GQA and MQA** We first compare with widely-used pretraining architectures, grouped-query attention (GQA) and multi-query attention (MQA) (Ainslie et al., 2023; Shazeer, 2019), which also aim to improve KV cache efficiency. While GQA and MQA reduce KV cache storage by sharing across heads, our method instead leverages recurrence to compress KV activations within dilation. Results in Fig. 5 demonstrate the competitive performance of RAT+. Notably, RAT+ provides greater flexibility: it requires training only a single dense model to support different levels of efficiency. It can also preserve the original KV cache size within local regions by configuring the local window size, which cannot be achieved by GQA.

*Table 7.* Accuracy on eight NIAH tasks from the RULER benchmark with 1024 KV entries per token. Note that top-k block methods do not reduce KV cache storage and can be sensitive to the block size D. Refer to Table 6 for their efficiency, Table 18, Table 17, Table 16 for more accuracy results.

| Method | S-1 | S-2 | S-3 | MK-1 | MK-2 | MK-3 | MV | MQ |
|---|---|---|---|---|---|---|---|---|
| **Attention** | | | | | | | | |
| Quest ($D^{\dagger} = 16$, K = 64) | 100.0 | 100.0 | 100.0 | 100.0 | 99.6 | 85.2 | 99.9 | 99.6 |
| MoBA ($D^{\dagger} = 16$, K = 64) | 100.0 | 99.8 | 96.2 | 99.6 | 97.8 | 77.0 | 95.4 | 96.5 |
| **RAT+** | | | | | | | | |
| SnapKV | 57.6 | 64.2 | 13.2 | 56.4 | 41.2 | 7.2 | 46.3 | 50.9 |
| StreamingLLM | 24.8 | 32.4 | 33.6 | 37.2 | 26.8 | 24.4 | 32.4 | 32.5 |
| Quest (D = 16, K = 64) | **100.0** | **100.0** | **100.0** | **100.0** | **100.0** | **100.0** | **99.2** | **100.0** |
| Quest (D = 64, K = 16) | 100.0 | 100.0 | 100.0 | 100.0 | 93.8 | 51.6 | 99.9 | 100.0 |
| Dilation (D = 4) | 100.0 | 100.0 | 97.8 | 99.8 | 97.4 | 88.8 | 99.8 | 99.5 |

**Comparison to top-k and SnapKV** For top-k block attention (e.g., Quest and MoBA), which does not reduce KV cache storage, we highlight in Subsec. 5.2 with ablations that recurrence also improves performance, achieving better results when applied to dense RAT+ compared to standard attention. We now also compare them with dilated attention. As shown in Table 7, the top-k block pattern can be sensitive to the block size, which also affects FLOPs overhead (e.g., block size 16 incurs $T/16$ additional FLOPs for block scoring) and contiguous memory access. Under the same number of KV entries per token, the best performance

reaches 99.2 with block size 16, but drops to 51.6 with block size 64. In comparison, dilated attention achieves 88.8, representing a strong result, especially given its extra advantage in reducing KV cache storage. Other KV cache reduction methods such as SnapKV (Li et al., 2024) show significantly worse performance, which is consistent with a recent hybrid design study (Feng et al., 2024).

**Orthogonality to top-k** Moreover, dilation can be used with the top-k block pattern in an orthogonal manner: top-k focuses on important blocks, while dilation represents the remaining ones. A similar design can also be found in Yuan et al. (2025), which trains an efficient architecture from scratch using MLPs to compress blocks. As shown in Table 19, combining the two can significantly improve top-k performance under the challenging setting $D = 64$, $K = 8$ (e.g., from 57.4 to 97.4 on NIAH-MK-2 and from 11.0 to 79.6 on NIAH-MK-3).

### 5.4. Scalability to 7B model scale

In this section, we further investigate scaling up to demonstrate that the proposed RAT+ architecture generalizes across different model sizes.

**Smaller loss gap between dense and sparse** Interestingly, as we scale the model from 1.3B to 2.6B and to 7.6B, the performance gap between sparse and dense attention (e.g., $D = 64$ and $D = 1$ in Fig. 6) becomes smaller, decreasing from 0.035 to 0.029 and then to 0.022. We attribute this to the increased capacity of recurrence with larger hidden dimensions, which better supports the required recurrence length such as 64.

**Downstream evaluation** Results in Table 8 further support the above observations. $D = 16$, $W = 256$ (an 8× re-

*Table 8.* **Summarized average performance of 7.6B RAT+.** Please see Table 21, Table 20 for more results.

| RAT+(7.6B) | CommonSense | LongBench | NIAH-S | NIAH-MK | NIAH-MV, MQ |
|---|---|---|---|---|---|
| D = 1 | **64.4** | 20.8 | 100.0 | 99.8 | 100.0 |
| D = 2 | 64.4 | 20.9 | 99.9 | 98.5 | 99.9 |
| D = 4 | 64.2 | **21.2** | 99.9 | 97.1 | 99.8 |
| D = 8 | 64.1 | 20.1 | 99.5 | 95.5 | 99.8 |
| D = 16 | 63.6 | 20.1 | 98.8 | 90.5 | 99.4 |
| D = 32 | 63.3 | 20.6 | 97.6 | 85.1 | 99.2 |
| D = 64 | 63.1 | 20.0 | 97.6 | 79.1 | 98.6 |
| D = 16, W = 256 | **64.7** | **21.1** | 99.6 | 93.9 | 99.6 |

duction in temporal-mixing FLOPs and KV cache) even outperforms dense attention on some tasks. On NIAH-MK-2, $D = 32$ achieves 78% accuracy at 1.5B, compared to 94.6% at 7.6B. We also observe that scaling to 7.6B does not improve the dense performance on the RBP task in LongBench, likely due to the limited amount of code data in the training set. We further provide comparisons with existing 7B models in Table 20, demonstrating the strong performance of our dense RAT+ at this scale.

### 5.5. Ablation studies

**Component ablation** We adopt simple recurrence for its simplicity, efficiency, and time-variant nature. We also ablate different usage schemes of Eq. (1) in Table 11. Moreover, Table 12 ablates the effect of our two proposed techniques. See Subsec. B.1 for more details.

**Choice of** $L = 64$**.** This is an empirical choice in RAT+. Intuitively, $L = 64$ is not a large sequence length, and recurrence can handle it reliably without suffering from memory degradation in long-context modeling. At the same time, it is not a small value: by covering dilation up to 64, it provides sufficient flexibility for choosing dilation sizes and constructing hybrid models with high speedups. Moreover, it is safer for length generalization compared to $L = 16$.

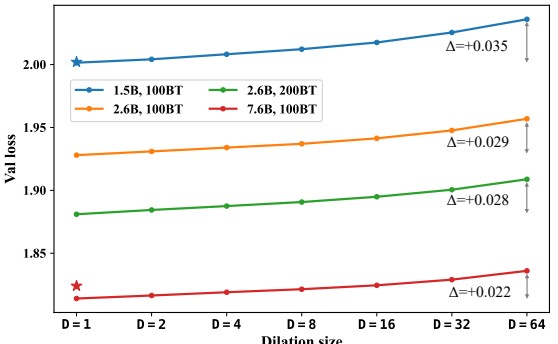

*Figure 6.* **Scaling-up experiments:** we report validation loss on a held-out 0.5B-token subset to illustrate the even smaller loss gap between dense and sparse variants as model size increases. The starred points refer to attention models trained with $D^\dagger = 1$ and $W = 64$, matched in training FLOPs at the same model scale, and are included as a reference for comparison with dense RAT+.

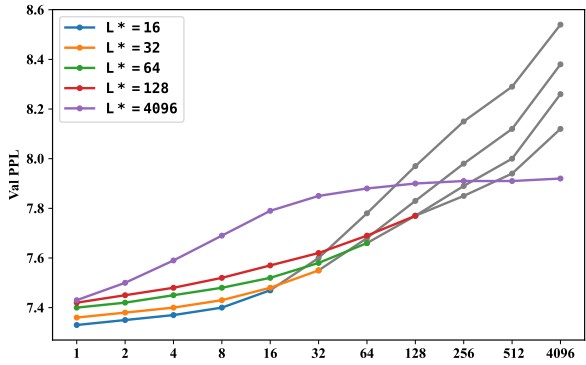

*Figure 7.* **Choice of** $L = 64$: $L^* = 16$ means $D = 1$ and $D = 16$ joint training to enable 16 length capability for recurrence. Different performance on $D = 1$ comes from different training FLOPs and different recurrence abilities in capturing the corresponding length. Grey points correspond to recurrence length generalization.

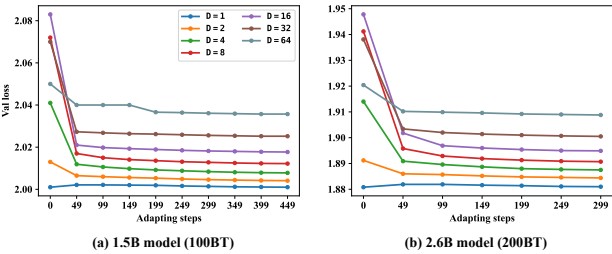

(a) 1.5B model (100BT)  (b) 2.6B model (200BT)

*Figure 8.* **1B-token adaptation on two pretrained models.** It is evident that various dilated patterns quickly achieve stable loss values within a few hundred million tokens. We employed a simple optimization scheme with no warmup, which may explain the slight loss increase of $D = 1$ at first, after which it recovers.

We also ablate other active recurrence lengths, as shown in Fig. 7. Taking these trade-offs and empirical results into account, we finally choose $L = 64$ in this paper.

**Adaptation analysis** We show in Fig. 8 that 1B tokens for resolution adaptation are sufficient to reach a stable loss for dilated attention in RAT+ models as well. Moreover, besides adapting each case individually, we also examine multiple-dilation adaptation in Table 10 within a single stage and find that it works well when covering several cases. Finally, although we have analyzed in Sec. 3 that the need for 1B-token adaptation arises from the attention mechanism itself, fully addressing this remains important for future work.

## 6. Related works

### 6.1. Efficient pretrained architectures

Many efficient architectures are designed and trained from scratch as alternatives to dense attention, including structured sparse patterns such as sliding-window and dilated attention (Ding et al., 2023; Beltagy et al., 2020; Zaheer et al., 2020; Hassani & Shi, 2022; Liu et al., 2021; Cohere et al., 2025; Yuan et al., 2025). Dilated-only schemes often require careful rate compositions to ensure full receptive fields and can be harder to optimize than dense models (Fournier et al., 2023; Hamaguchi et al., 2018). Other recent approaches build on state space models and linear attention (Gu & Dao, 2023; Dao & Gu, 2024; Yang et al., 2024; Zhang et al., 2024; Peng et al., 2023a; Katharopoulos et al., 2020; Sun et al., 2023; Orvieto et al., 2023), exploring the duality between linear recurrence and linear attention. A closely related architecture is RAT (Wei et al., 2025), which hierarchically combines recurrence and attention and can interpolate between them. Our analysis builds upon RAT, motivated by the observation that its inter-chunk attention mechanism is indeed dilated attention and yields strong training performance with recurrence. Finally, a key limitation of such efficient pretrained architectures is that they typically require training from scratch for each configura-

tion (e.g., dilation size in (Beltagy et al., 2020) or chunk size in RAT), which largely reduces inference-time flexibility across different task preferences and efficiency budgets.

### 6.2. Sparse inference for attention models

Many works also study inference-time sparsification of standard attention, motivated by the sparsity of attention maps. One direction emphasizes local attention (Xiao et al., 2023; Han et al., 2024) and its hybrids (Xiao et al., 2024b), while another identifies important tokens or blocks for each query token. Such importance-driven methods have been explored for both prefilling (Jiang et al., 2024; Lu et al., 2025; Lai et al., 2025) and decoding (Tang et al., 2024; Zhang et al., 2023; Lin et al., 2025; Xiao et al., 2024a; Zhang et al., 2025; Oren et al., 2024; Li et al., 2024). Liu et al. (2025) additionally applies continued pretraining to train a dedicated indexer. While sparsifying dense attention into these patterns is often effective, we explore the case of dilated attention, which dense pretrained models fail severely. We highlight it as an important structured pattern for its long-range connections beyond local-window schemes and its clean reductions in both FLOPs and KV cache size compared to most importance-driven approaches.

## 7. Conclusion and Future work

RAT+ points to a promising direction for improving inference efficiency: beyond investigating downstream methods, we can also design upstream architectures that support a broader set of effective sparse inference patterns. Large-scale experiments demonstrate the effectiveness of full-sequence recurrence and active recurrence learning for supporting dilated inference, highlighting RAT+'s potential for future language model development.

**Limitations and Future work** The efficiency of RAT+ could be further improved with dedicated CUDA kernels. Identifying optimal sparse configurations and further investigating the benefits from recurrence to other inference sparse patterns also present two interesting directions. RAT+ may also benefit several additional scenarios. First, it could be useful in tokenizer-free settings with byte-level input, where hierarchical representations are desirable. Dilation naturally controls attention granularity: early layers with small dilation capture fine-grained units such as bytes or words, while deeper layers with larger dilation model higher-level structures such as phrases or sentences more efficiently. Second, RAT+ may benefit parallel-style sampling for reasoning. For example, dilated attention can be used during the exploration phase for fast sampling, after which the model switches to the dense version to generate the final outputs with higher accuracy. This may be feasible because we found that the sparse and dense variants produce highly similar outputs by originating from the same dense checkpoint.

## Impact Statement

This paper presents work whose goal is to advance the field of Machine Learning by improving the efficiency of language models. There are many potential societal consequences of language modeling research, none of which we feel must be specifically highlighted here.

## Acknowledgements

Xiuying Wei's work is supported by the DVPS project, funded by the European Union's Horizon Europe Framework. We also sincerely thank the Swiss AI Initiative and the Swiss National Supercomputing Centre (CSCS) for supporting the computation through grants under project IDs a-infra01 and a109. We also thank Karin Getaz for administrative support, and the reviewers for their valuable suggestions and comments.

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

# A. Appendix

## A.1. Implementation details

**Implementation for Table 1.**   We largely follow the RAT implementation, using the same model architecture and training dataset but also some modifications. RAT shares linear projections for attention queries and keys across heads to match the parameter count of standard attention. In our implementation, we do not take this parameter sharing; instead, we keep dense linear projection for the recurrence and leave further lightweight considerations for future work. For the results in the third block, we simply adopt full-sequence recurrence and find that it improves performance.

**RAT+ pretraining**   For the 1.5B-parameter model, we use a model dimension of 2048, 24 Transformer layers, and a head dimension of 128, equipped with RMSNorm (Zhang & Sennrich, 2019) and no bias terms. The context window is set to 4096. We use standard RoPE (Su et al., 2024) instead of the inter-chunk RoPE used in RAT, for easier management of positional encoding, and put it after the recurrence function. The RoPE base is set to 10,000. Model parameters are initialized from a Gaussian distribution with a standard deviation of 0.02. We adopt the LLaMA2 tokenizer in all experiments. The optimization hyperparameters follow the rule described in Bi et al. (2024). We use a cosine-annealed learning rate schedule with a peak learning rate of $8.0 \times 10^{-4}$, decaying to $1.0 \times 10^{-6}$, and 5% warmup. The global batch size is set to 512.

For the 2.6B-parameter model trained on 100B tokens, we increase the model dimension to 2560 with 28 layers. All other hyperparameters are kept the same, except that we use a peak learning rate of $7.0 \times 10^{-4}$ and a global batch size of 640. For the 200B-token training setting, the peak learning rate remains $7.0 \times 10^{-4}$ with a global batch size of 768, following the same rule in Bi et al. (2024).

For the 7.6B-parameter model trained on 100B tokens, we adopt a model dimension of 4096, 32 Transformer layers, and a head dimension of 128. All other hyperparameters are kept the same, except that we set the peak learning rate to $6.0 \times 10^{-4}$ and the global batch size to 768.

**RAT+ adapting**   The adaptation process uses 1B randomly sampled tokens from the FineWeb-Edu dataset. In practice, we find that the training loss during adaptation stabilizes within a few hundred million tokens, as shown in Fig. 8. To ensure sufficient adaptation, we use 1B tokens in all experiments for safe results. We use the original batch size and a reduced learning rate ($0.1\times$ of the original). We later realize that a smaller learning rate (e.g., $2.0 \times 10^{-5}$) is theoretically more appropriate for the short adaptation, as it better preserves the original checkpoints. Since we observe almost no differences in evaluation results, we retain the original results for the 1.5B and 2.6B models. For the 7.6B model, we use $2.0 \times 10^{-5}$ as the adaptation learning rate.

**Evaluation**   Our evaluation setup largely follows the settings used in RAT, except for the NIAH tasks. We construct a dataset of approximately 7M tokens and apply a single-stage SFT before all evaluations. Compared to RAT, we sweep SFT for four epochs instead of one, to ensure that all models faithfully follow instructions.

**Efficiency**   The decoding implementation is straightforward, requiring only a single update for the recurrence and the use of the *SDPA* kernel in PyTorch. For training and prefilling, the recurrence is implemented using the *associative scan* in PyTorch. In the attention computation, we first use *flex attention* to handle the dilated connections by applying a dilation-level mask and returning the softmax denominator. We then compute the local and initial connections. The online softmax technique (Milakov & Gimelshein, 2018) is used to gather the results from the two parts. We expect that the efficiency of RAT+ can be further improved with dedicated CUDA kernel-level optimizations.

# B. Experiments

## B.1. Analyses and ablation studies

**Why recurrence?** We adopt recurrence primarily for its simplicity compared to MLPs, convolutions, attention, and linear attention. In contrast, these alternatives are heavier and more complex, which may introduce optimization difficulties in this hierarchical architecture. A second motivation is input-dependent temporal modeling. The recurrence employed here uses input-conditioned forget gates, which adapt to the dynamics of text sequences. MLPs are not time-variant, and both convolutions and attention require interactions with all tokens within the chunk, and thus would require more parameters under the overlapped chunk size.

For analyses, we include visualizations of the distributions at the initial recurrence steps (Fig. 9), head hit-rate analyses to support the claim that recurrence improves top-k block attention (Table 9), and multi-dilation adaptation results. For ablation studies, we present results on our two components in Table 12, and examine hierarchical recurrence in Table 11.

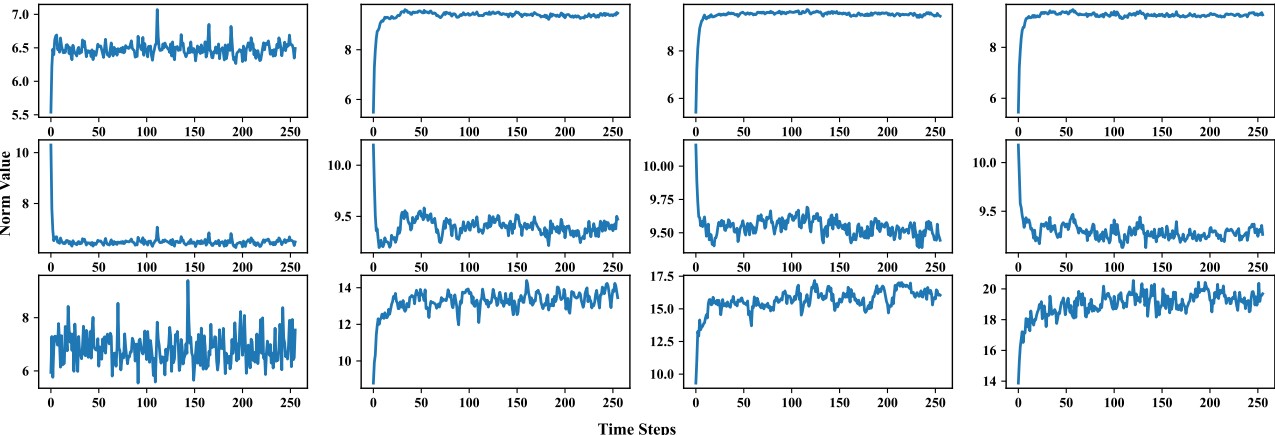

*Figure 9.* L2 norm values of recurrence outputs at different time steps. We observe that the outputs at early time steps differ significantly. The first row shows an initialized network using our simple recurrence at layers 0, 6, 18, and 23. The second row corresponds to the same initialized network but with a non-zero initial cell state provided to the recurrence. The third row shows the results of the pretrained network. We visualize the outputs of our simple recurrence here; a similar phenomenon has also been observed in standard recurrent networks, as reported in Cooijmans et al. (2016).

*Table 9.* **Top head hit rate analyses:** To further investigate why recurrence in RAT+ helps with top-k block attention, we measure the hit rate of each head, defined as whether the ground-truth token falls within the selected top-k blocks. These retrieval tasks require the model to locate a long digit or UUID among similar key-value pairs. Across 500 examples, a subset of heads consistently achieves high hit rates, consistent with prior findings (Wu et al., 2024) that only a few heads are critical for retrieval tasks. Compared to standard attention, top heads in RAT+ exhibit significantly higher hit rates, indicating improved block scoring. A more complete understanding of the underlying mechanisms is left to future work.

| Method | Acc | Top-1 | Top-2 | Top-3 | Top-4 | Top-5 |
|---|---|---|---|---|---|---|
| NIAH-MK-2 ($D = 64, K = 16$) | | | | | | |
|     Attention | 57.4 | 89.2 | 87.0 | 86.8 | 86.0 | 85.2 |
|     RAT+ | 93.8 | 97.0 | 93.8 | 93.2 | 93.2 | 90.8 |
| NIAH-MK-3 ($D = 16, K = 16$) | | | | | | |
|     Attention | 5.4 | 75.8 | 75.8 | 75.0 | 74.0 | 73.8 |
|     RAT+ | 94.8 | 99.8 | 99.6 | 99.6 | 99.6 | 97.6 |

*Table 10.* **Multi-dilation adaptation.** Compared to single-dilation adaptation, we also investigate whether multi-dilation adaptation can be effective. We jointly train on seven dilation choices during adaptation, and find that this performs well, achieving results comparable to the baseline where each dilation is adapted separately. However, we also observe that covering all possible dilation configurations within a single adaptation phase is highly challenging due to the extremely large hybrid space of $10^{20}$. This scale makes it difficult for the model to memorize each configuration, as it only performs $10^5$ forward passes, making it infeasible to encounter each configuration even once.

| Val PPL | $D = 1$ | $D = 2$ | $D = 4$ | $D = 8$ | $D = 16$ | $D = 32$ | $D = 64$ |
|---|---|---|---|---|---|---|---|
| single dilation adaptation | 7.40 | 7.42 | 7.44 | 7.48 | 7.52 | 7.58 | 7.66 |
| multiple dilation adaptation | 7.40 | 7.42 | 7.44 | 7.48 | 7.52 | 7.58 | 7.67 |

### B.2. Performance without positional encoding (NoPE)

While we use RoPE in the experiments above, we also investigate the NoPE setting. Surprisingly, the training curves remain stable under the same optimization hyperparameters and achieve comparable or slightly better perplexity when positional encodings are removed, as shown in Table 13. We hypothesize that this is enabled by joint training. NoPE assumes that

*Table 11.* **Hierarchical usage of recurrence.** PPL performance of a 200M RAT+ model on the PG19 book dataset with a context length of 8192. The first row shows strong performance across dilation settings. The second and third rows report additional design-choice ablations. As can be seen, removing recurrence over the attention keys leads to a slight increase in perplexity. We think this might be because the gated representations on the attention values can still be propagated across layers. In contrast, applying recurrence to the input of the *qkv* projections results in clearly worse performance, with $D = 4$ even higher than the $D = 128$ setting of the standard RAT+ variant.

| Model | Inference ($D$ value) | | | | | | | |
|---|---|---|---|---|---|---|---|---|
| | 1 | 2 | 4 | 8 | 16 | 32 | 64 | 128 |
| RAT+ ($L = T, L^* = 128$) | 10.71 | 10.87 | 10.97 | 11.05 | 11.15 | 11.20 | 11.28 | 11.39 |
| RAT+ (w/o recurrence on *key*) | 11.68 | 11.03 | 11.23 | 11.32 | 11.36 | 11.40 | 11.44 | 11.47 |
| Recurrence on input for *qkv* projections | 11.01 | 11.26 | 11.44 | 11.59 | 11.72 | 11.83 | 11.95 | 12.09 |

*Table 12.* **Ablation studies of RAT+.** All configurations are adapted from the corresponding pretrained models with 1B tokens, except for the w/o adaptation case. In the RAT+ ablation block: (1). the first row sets L equal to D at inference, and thus suffers from distribution shifts at the initial recurrence steps, leading to poor performance at $D = 1$; (2). The second row performs worse than RAT+ because dense connectivity is not sufficiently learned; (3). The third row removes active recurrence learning. Although dense performance is preserved, recurrence is not learned effectively, resulting in degraded performance at larger dilation sizes due to an incomplete receptive field; (4). The fourth row shows results without adaptation for our final variant. Note that even $D = 64$ still requires adaptation, since four initial tokens are additionally added at inference.

| Model | Train | Inference ($D$ value) | | | | | | |
|---|---|---|---|---|---|---|---|---|
| | | 1 | 2 | 4 | 8 | 16 | 32 | 64 |
| **RAT+** | | | | | | | | |
| $L = 64$ | $D = 1, D = 64$ | 7.4 | 7.42 | 7.45 | 7.48 | 7.52 | 7.58 | 7.67 |
| $L = T, L^* = 64$ | $D = 1, D = 64$ | 7.4 | 7.42 | 7.45 | 7.48 | 7.52 | 7.58 | 7.66 |
| **RAT+ ablations** | | | | | | | | |
| $L = 64$ ($L = D$ adapt) | $D = 1, D = 64$ | 8.39 | 8.14 | 7.84 | 7.68 | 7.6 | 7.6 | 7.66 |
| $L = 64$ | $0.5\mathcal{L}_{D=64} + 0.5\mathcal{L}_{D=1}$ | 7.52 | 7.56 | 7.58 | 7.62 | 7.66 | 7.73 | 7.82 |
| $L = T$ | $D = 1$ | 7.39 | 7.77 | 8.18 | 8.66 | 9.16 | 9.60 | 9.96 |
| $L = T, L^* = 64$ (w/o adapt) | $D = 1, D = 64$ | 7.4 | 7.49 | 7.69 | 7.94 | 8.03 | 7.93 | 7.77 |
| **attention baselines** | | | | | | | | |
| - | $D^\dagger = 1$ | 7.44 | 17.12 | 40.71 | 65.05 | 87.88 | 109.8 | 129.3 |
| - | $D^\dagger = 1, W = 64$ | 7.41 | 16.73 | 40.24 | 64.28 | 87.21 | 109.3 | 129.1 |
| - | $D^\dagger = 1, D^\dagger = 64$ | 7.82 | 16.22 | 34.74 | 52.96 | 70.94 | 90.1 | 109.9 |
| **RAT** ($L = 16$) | $D = 16$ | 8.41 | 8.08 | 7.79 | 7.64 | 7.61 | 7.74 | 7.93 |

*Table 13.* Validation perplexity of RAT+ when trained without positional encoding (NoPE). The validation perplexity (PPL) is comparable to, or slightly better than, the RoPE baseline.

| PE | Inference ($D$ value) | | | | | | |
|---|---|---|---|---|---|---|---|
| | 1 | 2 | 4 | 8 | 16 | 32 | 64 |
| **RoPE** | 7.40 | 7.42 | 7.45 | 7.48 | 7.52 | 7.58 | 7.66 |
| **NoPE** | 7.40 | 7.42 | 7.45 | 7.47 | 7.50 | 7.55 | 7.63 |

positional information can be implicitly inferred from causal masks and attention patterns (Kazemnejad et al., 2023). Then, in RAT+, with joint training, the sparse configuration $D = 64$ effectively reduces the attention span to 64 tokens under a context window of 4K. This shorter effective context makes it easier to leverage positional cues from the causal mask, compared to directly learning positional structure of 4K context length. In contrast, directly training attention-only models with full NoPE layers often results in slightly worse training loss, as also reported in prior work (Haviv et al., 2022; Yang et al., 2025).

### B.3. Detailed downstream evaluation results

**1.5B model scale** We provide detailed downstream evaluation results on the remaining LongBench tasks in Table 15, on NIAH tasks in Table 16, Table 17 and Table 18. In addition to comparative analyses of dilated and top-k block attention in the main text (e.g., sensitivity to block size and the help of recurrence in RAT+ for top-k attention), we further observe an interesting difference between them. Dilated attention finds NIAH-S-3 more challenging than NIAH-MK-2, NIAH-MQ, and NIAH-MV, exhibiting worse performance on NIAH-S-3. In contrast, top-k block attention performs better on NIAH-S-3 than on the other three tasks. We think this reflects the interaction between task characteristics and sparsification patterns. NIAH-S-3 and NIAH-MK-3 involve retrieving 32-digit UUID strings, where dilated attention with large dilation sizes may fail to capture the exact sequence; even a single incorrect digit leads to a complete loss of score. On the other hand, MQ, MV, and MK tasks involve multiple keys, queries, and values, which place a heavier burden on top-k block attention to identify the critical blocks, especially under less favorable configurations of $D$ and $K$.

*Table 14.* Detailed results of train-from-scratch models. Note that the dilated-only models fail in pretraining.

| Model (1.5B) | ARC-C | ARC-E | Hella. | LMB. | PIQA | Wino. | Avg. |
| --- | --- | --- | --- | --- | --- | --- | --- |
| | acc_n | acc | acc_n | acc | acc | acc | - |
| Dilated ($D^\dagger = 8$) | - | - | - | - | - | - | - |
| RAT ($L = 8, D = 8$) | 39.08 | 71.13 | 57.82 | 47.47 | 73.23 | 56.51 | 57.54 |
| Dilated ($D^\dagger = 16$) | - | - | - | - | - | - | - |
| RAT ($L = 16, D = 16$) | 39.76 | 72.6 | 56.95 | 46.03 | 72.14 | 54.46 | 56.99 |
| Dilated ($D^\dagger = 64, W = 64$) | 40.36 | 70.2 | 57.16 | 46.77 | 72.85 | 57.06 | 57.40 |
| RAT ($L = 64, D = 64, W = 64$) | 40.7 | 72.94 | 58.25 | 48.77 | 73.12 | 58.64 | 58.74 |
| attention | 40.1 | 71.84 | 58.5 | 49.95 | 72.42 | 57.14 | 58.33 |
| attention + recurrence | 40.02 | 72.9 | 58.92 | 49.51 | 73.29 | 58.41 | 58.84 |
| attention - ogate | 37.71 | 71.25 | 57.44 | 47.84 | 73.29 | 57.7 | 57.54 |
| attention - ogate + recurrence | 39.25 | 71.59 | 58.94 | 49.66 | 72.52 | 57.43 | 58.23 |

**7.6B model scale** Detailed results of 7.6B model scales are provided in Table 21 and Table 20.

*Table 15.* **Supplementary results on additional LongBench tasks of 1.5B models.** We omit these results from the main text as they are less representative. All models perform poorly on MusiQue and exhibit very similar performance on LCC. For the remaining summarization tasks, we observe that pretrained-only models tend to repeat prompts or outputs, which makes the quantitative results less reliable. Nevertheless, our dilated attention maintains performance close to the dense baseline, with the gap generally increasing as the dilation size becomes larger.

| RAT+ ($L^* = 64$) | Musique | GovReport | QMSUM | MultiNews | LCC |
|---|---|---|---|---|---|
| T = 4096 | | | | | |
| D = 1 | 7.2 | 15.34 | 17.56 | 18.25 | 19 |
| D = 2 | 7.85 | 16.13 | 17.47 | 16.51 | 19.1 |
| D = 4 | 7.59 | 15.45 | 17.22 | 18 | 19.69 |
| D = 8 | 7.81 | 13.98 | 16.6 | 15.37 | 19.1 |
| D = 16 | 7.61 | 15.1 | 16.36 | 16.08 | 18.37 |
| D = 32 | 6.92 | 12.45 | 15.6 | 14.75 | 17.96 |
| D = 64 | 6.85 | 12.47 | 15.17 | 16.07 | 18.26 |
| D = 128 | 6.9 | 11.82 | 15.06 | 15.48 | 18.8 |
| NTK extended | | | | | |
| T = 16384 | | | | | |
| D = 1 | 7.62 | 21.87 | 17.86 | 18.45 | 18.81 |
| D = 4 | 7.44 | 21.72 | 17.97 | 19.56 | 18.55 |

*Table 16.* **Retrieval ability**: Accuracy performance with exact match scoring on the Needle-in-Haystack tasks with different configurations from the RULER benchmark (Hsieh et al., 2024). We use T = 4096 and bold less favorable results. We found they mainly occur when the text query/key/values are the UUID, and may incorrectly copy parts of the answers. Also, in NIAH-MK-2, larger dilation sizes also perform worse.

| SFT models | NIAH-S-1 | NIAH-S-2 | NIAH-S-3 | NIAH-MK-1 | NIAH-MK-2 | NIAH-MK-3 | NIAH-MV | NIAH-MQ |
|---|---|---|---|---|---|---|---|---|
| D = 1 | 100 | 100 | 100 | 100 | 100 | 99.6 | 99.85 | 100 |
| D = 2 | 100 | 100 | 99.2 | 100 | 99.2 | 96 | 99.8 | 99.8 |
| D = 4 | 100 | 100 | 97.8 | 99.8 | 97.4 | **88.8** | 99.75 | 99.5 |
| D = 8 | 100 | 100 | 96 | 99.8 | 96.4 | **73.8** | 99.4 | 99.15 |
| D = 16 | 100 | 100 | **89.2** | 99.6 | 95.2 | **54.6** | 99.3 | 98.15 |
| D = 32 | 100 | 100 | **78.8** | 99.2 | **78** | **40.4** | 97.35 | 96.35 |
| D = 64 | 100 | 100 | **76** | 98.4 | **46.8** | **30** | 97.35 | 93.9 |

*Table 17.* Retrieval ability of the top-k block attention for different models. On the following three pretrained-plus-SFT models, we apply Quest (Tang et al., 2024) style top-k block attention, where the critical block is determined by the Min/Max statistics of attention keys within each block.

| Quest top-k | NIAH-S-1 | NIAH-S-2 | NIAH-S-3 | NIAH-MK-1 | NIAH-MK-2 | NIAH-MK-3 | NIAH-MV | NIAH-MQ |
|---|---|---|---|---|---|---|---|---|
| **attention** | | | | | | | | |
| $D^\dagger = 1$, All | 100 | 100 | 99.8 | 100 | 100 | 99.6 | 100 | 99.9 |
| $D^\dagger = 16$, K = 16 | 100 | 99.8 | 99.4 | 99.6 | **71.2** | **5.4** | 93.95 | 94.25 |
| $D^\dagger = 16$, K = 64 | 100 | 100 | 100 | 100 | 99.6 | **85.2** | 99.9 | 99.6 |
| $D^\dagger = 64$, K = 8 | 100 | 100 | **85.8** | 98.4 | **20.6** | **0.2** | **87.55** | **88.55** |
| $D^\dagger = 64$, K = 16 | 100 | 100 | 99 | 99.8 | **63.2** | **15.2** | 97.45 | 96.85 |
| **RAT+** | | | | | | | | |
| D = 1, All | 100 | 100 | 100 | 100 | 100 | 99.6 | 99.85 | 100 |
| D = 16, K = 16 | 100 | 100 | 100 | 100 | 97.6 | 94.8 | 99.85 | 99.9 |
| D = 16, K = 64 | 100 | 100 | 100 | 100 | 100 | 99.2 | 100 | 99.95 |
| D = 64, K = 8 | 100 | 100 | 99.8 | 100 | **57.4** | **11** | 98.75 | 99.7 |
| D = 64, K = 16 | 100 | 100 | 100 | 100 | 93.8 | **51.6** | 99.85 | 99.95 |
| **RAT+ (no ARL in SFT)** | | | | | | | | |
| D = 1, All | 100 | 100 | 100 | 100 | 99.4 | 98.8 | 99.8 | 99.85 |
| D = 16, K = 16 | 100 | 100 | 100 | 100 | **84.2** | **27.4** | 96.95 | 99.4 |
| D = 16, K = 64 | 100 | 100 | 100 | 100 | 99.2 | 97.6 | 99.7 | 99.85 |
| D = 64, K = 8 | 100 | 100 | 100 | 99.8 | **32** | **5.4** | 90.25 | 98.55 |
| D = 64, K = 16 | 100 | 100 | 100 | 100 | **76.8** | **28** | 98.05 | 99.6 |

*Table 18.* Retrieval ability of top-k block attention across different models. For the following three pretrained-plus-SFT models, we apply MoBA-style top-k block attention (Lu et al., 2025), where the critical block is selected based on the mean-pooled attention keys within each block. Notably, we apply this mechanism to all layers and to both the prefilling and decoding stages, without any further fine-tuning. In contrast, the original MoBA paper uses continued pretraining and reports applying the method only during the prefilling stage to avoid noticeable accuracy degradation. Our goal here is primarily to demonstrate the effectiveness of RAT+ over the attention backbone.

| MoBA top-k | NIAH-S-1 | NIAH-S-2 | NIAH-S-3 | NIAH-MK-1 | NIAH-MK-2 | NIAH-MK-3 | NIAH-MV | NIAH-MQ |
|---|---|---|---|---|---|---|---|---|
| **attention** | | | | | | | | |
| $D^\dagger = 1$, All | 100 | 100 | 99.8 | 100 | 100 | 99.6 | 100 | 99.9 |
| $D^\dagger = 16, K = 16$ | 100 | 91 | **56** | **84** | **65.2** | **6.2** | **48.1** | **55.9** |
| $D^\dagger = 16, K = 64$ | 100 | 99.8 | 96.2 | 99.6 | 97.8 | **77** | 95.35 | 96.45 |
| $D^\dagger = 64, K = 8$ | 90.2 | **66.8** | **20** | **54.2** | **29.8** | **5.8** | **30.1** | **34.55** |
| $D^\dagger = 64, K = 16$ | 99.2 | 95.8 | **79** | 91.8 | **66.2** | **22.6** | **70.7** | **76.35** |
| **RAT+** | | | | | | | | |
| $D = 1$, All | 100 | 100 | 100 | 100 | 100 | 99.6 | 99.85 | 100 |
| $D = 16, K = 16$ | 100 | 100 | 97.8 | 100 | 96.4 | **86.8** | 99.15 | 99.7 |
| $D = 16, K = 64$ | 100 | 100 | 99.8 | 100 | 99.8 | 99.2 | 100 | 99.95 |
| $D = 64, K = 8$ | 98.4 | 99.6 | **79.6** | 95.4 | **39.4** | **10.8** | 81.55 | **88.1** |
| $D = 64, K = 16$ | 99.8 | 100 | 96.8 | 99.8 | **80.2** | **40.2** | 96.55 | 98.2 |

*Table 19.* **Orthogonality between dilated and top-k block attention.** Performance of combining dilated and top-k block attention (e.g., $D = 64 \mid D = 64, K = 16$), where each token attends to every token within $K$ critical blocks, while also attending to other blocks through compressed information provided by dilation. Here we focus on the challenging setting with a large block size of 64 for the top-k block pattern.

| Model | NIAH-MK-2 | NIAH-MK-3 |
|---|---|---|
| $D = 64, K = 8$ | 57.4 | 11.0 |
| $D = 64 \mid D = 64, K = 8$ | **97.4** | **79.6** |
| $D = 64, K = 16$ | 93.8 | 51.6 |
| $D = 64 \mid D = 64, K = 16$ | **99.2** | **97.0** |

*Table 20.* **Comparison with existing 7B models trained on a similar data scale.** Note that different models are trained on different data sources, which may lead to variations in downstream performance. Models are from De et al. (2024); Biderman et al. (2023).

| Model | ARC-C | ARC-E | Hella. | LMB. | PIQA | Wino. |
|---|---|---|---|---|---|---|
| - | acc_n | acc/acc_n | acc_n | acc | acc/acc_n | acc |
| Griffin (300BT) | 47.9 | -/75.4 | 78.6 | - | -/81.0 | 72.6 |
| Pythia (300BT) | 35.2 | 67.3/61.2 | 63.9 | 67.2 | 75.1/76.2 | 61.3 |
| Pythia (100BT) | 29.3 | 62.3/56.0 | 56.4 | 61.2 | 72.4/73.6 | 56.8 |
| RAT+ (100BT) | 47.3 | 77.6/76.0 | 68.2 | 56.6 | 75.2/77.2 | 61.7 |

*Table 21.* **Accuracy performance of 7.6B RAT+ on LongBench and needle-in-haystack tasks.**

| 7.6B RAT+ | NQA | Qasper | MF | HQA | 2WQA | RBP | Avg. | S-1 | S-2 | S-3 | MK-1 | MK-2 | MK-3 | MV | MQ |
|---|---|---|---|---|---|---|---|---|---|---|---|---|---|---|---|
| D = 1 | 16.07 | 17.24 | 29.25 | 16.49 | 19.79 | 25.89 | 20.8 | 100 | 100 | 100 | 100 | 99.8 | 99.4 | 100 | 99.95 |
| D = 2 | 16.2 | 16.84 | 30.22 | 17.89 | 18.77 | 25.45 | 20.9 | 100 | 100 | 99.6 | 100 | 99.4 | 96 | 99.9 | 99.85 |
| D = 4 | 15.4 | 17.01 | 31.51 | 17.76 | 19.36 | 26.24 | 21.2 | 100 | 100 | 99.6 | 99.8 | 99.8 | 91.8 | 99.9 | 99.6 |
| D = 8 | 15.87 | 16.1 | 28.15 | 16.11 | 18.03 | 26.3 | 20.1 | 100 | 100 | 98.6 | 100 | 99 | 87.6 | 99.85 | 99.65 |
| D = 16 | 15.53 | 16.3 | 28.86 | 16.28 | 19.36 | 24.43 | 20.1 | 99.8 | 100 | 96.6 | 99.8 | 97.8 | 74 | 99.55 | 99.15 |
| D = 32 | 15.56 | 15.93 | 28.23 | 17.21 | 21.48 | 25.23 | 20.6 | 100 | 100 | 92.8 | 99.8 | 94.6 | 61 | 99.7 | 98.75 |
| D = 64 | 15.58 | 15.77 | 26.95 | 16.35 | 19.35 | 26 | 20.0 | 100 | 100 | 92.8 | 99.8 | 85.6 | 51.8 | 99.35 | 97.9 |
| D = 16,W = 256 | 16.06 | 15.94 | 28.62 | 18.03 | 19.9 | 28.34 | 21.1 | 100 | 100 | 98.8 | 100 | 98.4 | 83.4 | 99.65 | 99.5 |

## B.4. Additional efficiency results

In this part, we provide additional detailed efficiency results in Table 22, Table 23, Table 24, and Table 25. We also highlight several speedups in the tables.

*Table 22.* Temporal-mixing operator and block (linear projections included) prefilling time across different sequence lengths for hidden dimension 2048. The latency (ms) is tested on 262K sequences of tokens. $D^{\dagger} = 1$ means the attention operator or block without the recurrence.

| Latency (H=2048) | 4K | 8K | 16K | 32K | 65K | 131K | 262K |
|---|---|---|---|---|---|---|---|
| **Temporal-mixing operator** | | | | | | | |
| $D^{\dagger} = 1$ | 16.3 | 28.5 | 51.8 | 101.1 | 197.9 | 422.3 | 897.8 |
| $D = 1$ | 32.0 | 43.9 | 67.8 | 118.2 | 218.4 | 445.3 | 942.4 |
| $D = 2$ | 30.9 | 37.9 | 51.9 | 83.3 | 144.2 | 280.1 | 587.5 |
| $D = 4$ | 26.9 | 30.6 | 37.7 | 54.2 | 87.6 | 165.4 | 330.7 |
| $D = 8$ | 24.3 | 25.9 | 29.1 | 37.7 | 54.4 | 98.7 | 190.9 |
| $D = 16$ | 23.3 | 24.3 | 26.1 | 31.4 | 42.8 | 74.5 | 143.6 |
| $D = 32$ | 23.1 | 23.5 | 24.3 | 28.3 | 36.3 | 62.0 | 119.4 |
| $D = 64$ | 22.8 | 23.3 | 23.7 | 26.8 | 33.6 | 56.2 | 107.9 |
| $D = 8, W = 512$ | 27.2 | 29.1 | 32.0 | 40.2 | 58.0 | 101.8 | 199.4 |
| $D = 16, W = 256$ | 25.2 | 26.1 | 27.9 | 33.3 | 44.3 | 76.5 | 148.6 |
| $D^{\dagger} = 1 / D = 16$ | 0.7× | 1.2 × | 2.0× | 3.2× | 4.6× | 5.7× | 6.3× |
| $D = 1 / D = 16$ | 1.4× | 1.8 × | 2.6× | 3.8 × | 5.1× | 6.0× | 6.6× |
| **Temporal-mixing block** | | | | | | | |
| $D^{\dagger} = 1$ | 40.3 | 54.5 | 78.4 | 125.9 | 227.3 | 441.9 | 936.6 |
| $D = 1$ | 59.9 | 75.5 | 98.0 | 149.5 | 248.5 | 464.8 | 972.7 |
| $D = 2$ | 59.4 | 67.2 | 83.1 | 115.7 | 182.3 | 316.5 | 614.2 |
| $D = 4$ | 54.6 | 57.5 | 67.9 | 85.5 | 120.3 | 198.7 | 359.7 |
| $D = 8$ | 51.9 | 53.4 | 57.0 | 65.4 | 82.5 | 121.0 | 221.2 |
| $D = 16$ | 48.8 | 51.6 | 53.8 | 59.8 | 72.5 | 102.5 | 177.0 |
| $D = 32$ | 49.6 | 50.7 | 51.4 | 55.8 | 61.6 | 89.2 | 146.6 |
| $D = 64$ | 50.3 | 50.4 | 50.6 | 51.0 | 61.6 | 79.5 | 133.7 |
| $D = 8, W = 512$ | 53.5 | 56.7 | 60.8 | 66.3 | 88.2 | 126.0 | 223.7 |
| $D = 16, W = 256$ | 53.0 | 53.5 | 55.3 | 60.7 | 71.7 | 103.2 | 179.5 |
| $D^{\dagger} = 1 / D = 16$ | 0.8× | 1.1 × | 1.5× | 2.1× | 3.1× | 4.3× | 5.3× |
| $D = 1 / D = 16$ | 1.2× | 1.5 × | 1.8× | 2.5× | 3.4× | 4.5× | 5.5× |

*Table 23.* Decoding latency of temporal-mixing operator and blocks for hidden dimension 2048. The latency (ms) is tested on generating batches of tokens with $B = 128$, $B = 256$, $B = 512$ at specified positions.

| Latency (H=2048) | 4K | 8K | 16K | 32K | 65K | 131K | 262K |
|---|---|---|---|---|---|---|---|
| **Temporal-mixing operator** | | | | | | | |
| $B = 128$ | | | | | | | |
| D$^\dagger$ = 1 | 1.19 | 2.33 | 4.62 | 9.18 | 18.34 | OOM | OOM |
| D = 1 | 1.24 | 2.38 | 4.67 | 9.24 | 18.38 | OOM | OOM |
| D = 2 (vs. D$^\dagger$ = 1) | 0.69 (1.7×) | 1.26 (1.8×) | 2.37 (1.9×) | 4.66 (2.1×) | 9.22 (2.0×) | 18.35 | OOM |
| D = 4 | 0.49 | 0.69 | 1.26 | 2.37 | 4.66 | 9.23 | 18.35 |
| D = 8 | 0.47 | 0.58 | 0.69 | 1.26 | 2.37 | 4.66 | 9.22 |
| D = 16 | 0.45 | 0.43 | 0.47 | 0.69 | 1.26 | 2.38 | 4.66 |
| D = 32 | 0.39 | 0.41 | 0.45 | 0.59 | 0.69 | 1.26 | 2.37 |
| D = 64 | 0.36 | 0.48 | 0.44 | 0.45 | 0.48 | 0.69 | 1.26 |
| D = 8, W = 512 | 0.55 | 0.62 | 0.82 | 1.41 | 2.59 | 4.91 | 9.63 |
| D = 16, W = 256 | 0.48 | 0.5 | 0.54 | 0.76 | 1.35 | 2.51 | 4.85 |
| $B = 256$ | | | | | | | |
| D$^\dagger$ = 1 | 2.32 | 4.58 | 9.08 | 18.07 | OOM | OOM | OOM |
| D = 1 | 2.38 | 4.64 | 9.14 | 18.14 | OOM | OOM | OOM |
| D = 2 (vs. D$^\dagger$ = 1) | 1.27 (1.8×) | 2.41 (1.9×) | 4.63 (2.0×) | 9.13 | 18.13 | OOM | OOM |
| D = 4 | 0.7 | 1.27 | 2.41 | 4.63 | 9.13 | 18.13 | OOM |
| D = 8 | 0.49 | 0.7 | 1.27 | 2.41 | 4.63 | 9.13 | 18.13 |
| D = 16 (vs. D$^\dagger$ = 1) | 0.44 (5.3×) | 0.55 (8.3×) | 0.7 (13.0×) | 1.27 (14.2×) | 2.41 | 4.63 | 9.14 |
| D = 32 | 0.42 | 0.55 | 0.49 | 0.7 | 1.27 | 2.41 | 4.64 |
| D = 64 | 0.38 | 0.47 | 0.5 | 0.49 | 0.7 | 1.27 | 2.41 |
| D = 8, W = 512 | 0.68 | 0.99 | 1.54 | 2.72 | 5.03 | 9.66 | 18.92 |
| D = 16, W = 256 | 0.63 | 0.56 | 0.84 | 1.42 | 2.58 | 4.9 | 9.53 |
| $B = 512$ | | | | | | | |
| D$^\dagger$ = 1 | 4.59 | 9.09 | 18.08 | OOM | OOM | OOM | OOM |
| D = 1 | 4.65 | 9.15 | 18.14 | OOM | OOM | OOM | OOM |
| D = 2 | 2.43 | 4.71 | 9.15 | 18.15 | OOM | OOM | OOM |
| D = 4 | 1.29 | 2.43 | 4.71 | 9.16 | 18.14 | OOM | OOM |
| D = 8 | 0.72 | 1.29 | 2.43 | 4.72 | 9.15 | 18.14 | OOM |
| D = 16 (vs. D$^\dagger$ = 1) | 0.45 (10.2×) | 0.72 (12.6×) | 1.29 (14.0×) | 2.43 | 4.71 | 9.16 | 18.15 |
| D = 32 | 0.49 | 0.45 | 0.72 | 1.29 | 2.43 | 4.71 | 9.15 |
| D = 64 (vs. D = 16) | 0.46 (1.0×) | 0.47 (1.5×) | 0.51 (2.5×) | 0.72 (3.4×) | 1.29 (3.7×) | 2.43 (3.8×) | 4.71 (3.9×) |
| D = 8, W = 512 | 1.25 | 1.86 | 2.96 | 5.3 | 9.95 | 19.23 | OOM |
| D = 16, W = 256 | 0.71 | 0.99 | 1.57 | 2.73 | 5.05 | 9.68 | 18.91 |
| **Temporal-mixing block** | | | | | | | |
| $B = 512$ | | | | | | | |
| D$^\dagger$ = 1 | 4.72 | 9.21 | 18.22 | OOM | OOM | OOM | OOM |
| D = 1 | 4.8 | 9.29 | 18.29 | OOM | OOM | OOM | OOM |
| D = 2 (vs. D$^\dagger$ = 1) | 2.58 (1.8×) | 4.86 (1.9×) | 9.31 (2.0×) | 18.3 | OOM | OOM | OOM |
| D = 4 (vs. D$^\dagger$ = 1) | 1.44 (3.3×) | 2.58 (3.6×) | 4.86 (3.7×) | 9.31 | 18.3 | OOM | OOM |
| D = 8 | 1 | 1.45 | 2.58 | 4.87 | 9.31 | 18.29 | OOM |
| D = 16 (vs. D$^\dagger$ = 1) | 0.88 (5.4×) | 1.13 (8.2×) | 1.45 (12.6×) | 2.59 | 4.87 | 9.32 | 18.31 |
| D = 32 | 0.83 | 0.93 | 1.06 | 1.45 | 2.59 | 4.87 | 9.32 |
| D = 64 | 0.76 | 0.77 | 0.92 | 1.01 | 1.45 | 2.59 | 4.87 |
| D = 8, W = 512 | 1.4 | 2.01 | 3.11 | 5.46 | 10.1 | 19.38 | OOM |
| D = 16, W = 256 | 0.98 | 1.18 | 1.72 | 2.88 | 5.2 | 9.84 | 19.08 |

*Table 24.* Temporal-mixing operator and block (linear projections included) prefilling time across different sequence lengths for hidden dimension 4096. The latency (ms) is tested on 262K tokens. We observe even better speed-up on both operator and block levels compared to H=2048.

| Latency (H=4096) | 4K | 8K | 16K | 32K | 65K | 131K | 262K |
|---|---|---|---|---|---|---|---|
| **Temporal-mixing operator** | | | | | | | |
| $D^\dagger = 1$ | 33.23 | 56.90 | 106.03 | 203.94 | 397.90 | 839.06 | 1789.60 |
| $D = 1$ | 64.36 | 88.78 | 138.34 | 237.04 | 429.17 | 875.68 | 1839.85 |
| $D = 2$ | 61.77 | 76.53 | 104.56 | 165.74 | 281.48 | 530.33 | 1100.72 |
| $D = 4$ | 53.81 | 61.16 | 75.78 | 105.75 | 168.60 | 296.81 | 583.40 |
| $D = 8$ | 48.67 | 52.05 | 58.28 | 72.58 | 102.40 | 165.94 | 303.43 |
| $D = 16$ | 46.89 | 48.92 | 52.41 | 59.84 | 77.55 | 117.82 | 211.29 |
| $D = 32$ | 46.50 | 47.40 | 49.12 | 53.01 | 64.75 | 92.56 | 161.59 |
| $D = 64$ | 45.81 | 46.99 | 47.76 | 50.02 | 58.52 | 80.46 | 136.50 |
| $D = 8, W = 512$ | 54.21 | 57.21 | 64.32 | 77.57 | 108.21 | 171.87 | 313.64 |
| $D = 16, W = 256$ | 50.32 | 52.03 | 55.85 | 62.88 | 82.04 | 119.74 | 214.41 |
| $D = 1 / D = 16$ | 1.37× | 1.81× | 2.64× | 3.96× | 5.53× | 7.43× | 8.71× |
| $D^\dagger = 1 / D = 16$ | 0.71× | 1.16× | 2.02× | 3.41× | 5.13× | 7.12× | 8.47× |
| **Temporal-mixing block** | | | | | | | |
| $D^\dagger = 1$ | 116.53 | 141.30 | 187.46 | 293.98 | 479.93 | 913.62 | 1877.32 |
| $D = 1$ | 161.27 | 185.54 | 234.56 | 333.38 | 535.64 | 981.81 | 1960.33 |
| $D = 2$ | 159.13 | 179.87 | 200.92 | 273.59 | 379.28 | 635.22 | 1204.32 |
| $D = 4$ | 152.13 | 159.08 | 177.93 | 212.50 | 272.70 | 408.52 | 694.38 |
| $D = 8$ | 146.65 | 150.38 | 157.87 | 169.13 | 202.16 | 265.14 | 417.62 |
| $D = 16$ | 146.54 | 146.11 | 151.00 | 158.90 | 179.45 | 219.13 | 313.70 |
| $D = 32$ | 144.17 | 145.05 | 143.80 | 151.24 | 164.32 | 190.89 | 261.45 |
| $D = 64$ | 142.80 | 144.58 | 145.22 | 145.94 | 159.37 | 178.21 | 236.68 |
| $D = 8, W = 512$ | 151.38 | 160.30 | 164.59 | 175.73 | 211.21 | 272.00 | 419.63 |
| $D = 16, W = 256$ | 147.94 | 148.05 | 152.04 | 161.56 | 181.04 | 219.94 | 316.19 |
| $D = 1 / D = 16$ | 1.10× | 1.27× | 1.55× | 2.10× | 2.98× | 4.48× | 6.25× |
| $D^\dagger = 1 / D = 16$ | 0.80× | 0.97× | 1.24× | 1.85× | 2.67× | 4.17× | 5.98× |

*Table 25.* Decoding latency of temporal-mixing operator and blocks for hidden dimension 4096. The latency (ms) is tested on generating batches of tokens with $B = 128$, $B = 256$, $B = 512$ at specified positions.

| Latency (H=4096) | 4K | 8K | 16K | 32K | 65K | 131K | 262K |
|---|---|---|---|---|---|---|---|
| **Temporal-mixing operator** | | | | | | | |
| $B = 128$ | | | | | | | |
| $D^\dagger = 1$ | 2.32 | 4.58 | 9.08 | 18.09 | OOM | OOM | OOM |
| $D = 1$ | 2.38 | 4.64 | 9.14 | 18.14 | OOM | OOM | OOM |
| $D = 2$ (vs. $D^\dagger = 1$) | 1.27 (1.8×) | 2.42 (1.9×) | 4.64 (2.0×) | 9.14 (2.0×) | 18.17 | OOM | OOM |
| $D = 4$ | 0.70 | 1.27 | 2.41 | 4.64 | 9.13 | 18.15 | OOM |
| $D = 8$ | 0.49 | 0.70 | 1.28 | 2.42 | 4.64 | 9.14 | 18.15 |
| $D = 16$ | 0.41 | 0.48 | 0.71 | 1.28 | 2.42 | 4.64 | 9.15 |
| $D = 32$ | 0.54 | 0.59 | 0.53 | 0.71 | 1.28 | 2.42 | 4.64 |
| $D = 64$ | 0.43 | 0.55 | 0.46 | 0.58 | 0.70 | 1.27 | 2.41 |
| $D = 8, W = 512$ | 0.68 | 0.98 | 1.54 | 2.72 | 5.04 | 9.67 | 18.97 |
| $D = 16, W = 256$ | 0.61 | 0.56 | 0.84 | 1.42 | 2.58 | 4.91 | 9.53 |
| $B = 256$ | | | | | | | |
| $D^\dagger = 1$ | 4.59 | 9.09 | 18.09 | OOM | OOM | OOM | OOM |
| $D = 1$ | 4.65 | 9.15 | 18.15 | OOM | OOM | OOM | OOM |
| $D = 2$ | 2.43 | 4.72 | 9.15 | 18.15 | OOM | OOM | OOM |
| $D = 4$ | 1.29 | 2.43 | 4.71 | 9.15 | 18.15 | OOM | OOM |
| $D = 8$ | 0.73 | 1.30 | 2.44 | 4.72 | 9.17 | 18.16 | OOM |
| $D = 16$ (vs. $D^\dagger = 16$) | 0.45 (10.2×) | 0.73 (12.5×) | 1.30 (14.0×) | 2.44 | 4.72 | 9.16 | 18.15 |
| $D = 32$ | 0.48 | 0.50 | 0.73 | 1.30 | 2.44 | 4.72 | 9.16 |
| $D = 64$ | 0.45 | 0.55 | 0.48 | 0.72 | 1.29 | 2.44 | 4.71 |
| $D = 8, W = 512$ | 1.25 | 1.88 | 2.96 | 5.30 | 9.95 | 19.23 | OOM |
| $D = 16, W = 256$ | 0.71 | 0.99 | 1.57 | 2.73 | 5.06 | 9.69 | 18.92 |
| $B = 512$ | | | | | | | |
| $D^\dagger = 1$ | 9.11 | 18.10 | OOM | OOM | OOM | OOM | OOM |
| $D = 1$ | 9.21 | 18.21 | OOM | OOM | OOM | OOM | OOM |
| $D = 2$ | 4.79 | 9.36 | 18.22 | OOM | OOM | OOM | OOM |
| $D = 4$ | 2.50 | 4.79 | 9.35 | 18.22 | OOM | OOM | OOM |
| $D = 8$ | 1.37 | 2.52 | 4.80 | 9.37 | 18.22 | OOM | OOM |
| $D = 16$ | 0.81 | 1.37 | 2.51 | 4.80 | 9.36 | 18.22 | OOM |
| $D = 32$ | 0.60 | 0.81 | 1.37 | 2.51 | 4.80 | 9.37 | 18.23 |
| $D = 64$ | 0.54 | 0.60 | 0.81 | 1.37 | 2.51 | 4.79 | 9.35 |
| $D = 8, W = 512$ | 2.41 | 3.69 | 5.84 | 10.53 | 19.78 | OOM | OOM |
| $D = 16, W = 256$ | 1.35 | 1.90 | 3.06 | 5.38 | 10.02 | 19.29 | OOM |
| **Temporal-mixing block** | | | | | | | |
| $B = 512$ | | | | | | | |
| $D^\dagger = 1$ | 9.46 | 18.46 | OOM | OOM | OOM | OOM | OOM |
| $D = 1$ | 9.62 | 18.63 | OOM | OOM | OOM | OOM | OOM |
| $D = 2$ (vs. $D^\dagger = 1$) | 5.19 (1.8×) | 9.78 (1.9×) | 18.64 | OOM | OOM | OOM | OOM |
| $D = 4$ | 2.91 | 5.21 | 9.77 | 18.64 | OOM | OOM | OOM |
| $D = 8$ | 1.77 | 2.91 | 5.20 | 9.76 | 18.63 | OOM | OOM |
| $D = 16$ | 1.21 | 1.78 | 2.91 | 5.20 | 9.77 | 18.63 | OOM |
| $D = 32$ | 1.01 | 1.21 | 1.77 | 2.92 | 5.20 | 9.76 | 18.63 |
| $D = 64$ | 0.96 | 1.03 | 1.22 | 1.78 | 2.92 | 5.20 | 9.78 |
| $D = 8, W = 512$ | 2.81 | 4.10 | 6.24 | 10.91 | 20.17 | OOM | OOM |
| $D = 16, W = 256$ | 1.76 | 2.31 | 3.48 | 5.79 | 10.43 | 19.70 | OOM |

