# OpenReview forum: "RAT+: Train Dense, Infer Sparse - Recurrence Augmented Attention for Dilated Inference"
_ICML.cc/2026/Conference — ICML 2026 regular_

### Official Review · Reviewer_j5q7 · 2026-03-02

**Soundness:** 3
**Presentation:** 2
**Significance:** 2
**Originality:** 2
**Overall Recommendation:** 4
**Confidence:** 4

**Summary:**

The paper proposes RAT+, a dense-pretraining architecture that leverages Recurrence Augmented Dilated Attention for efficient long sequence inference. RAT+ decouples the recurrent and dilated attention group, enabling fast adaption from pre-trained models. RAT+ closely matches the dense model accuracy at $D=16$ dilated attention and outperforms top-k block attention.

**Compliance With Llm Reviewing Policy:**

Affirmed.

**Final Justification:**

My score keeps as a weak accept. The paper gives an architecture which gives fundamental changes on the attention mechanism. However, the whole context is based on lossy acceleration. The motivation is doubtful, but the performance looks fine.

**Key Questions For Authors:**

1. You mention about the top-k block attention methods. Actually, the sparse attention methods are not orthogonal to your method. When we consider the decoding phase, the block retrieval + attention with dilated window also achieves comparable accleration. Could you summarize the motivation and core advantage to the sparse attention variants?

**Limitations:**

Limitaions are not discussed in the paper.

**Strengths And Weaknesses:**

Soundness: There are a wide range of evaluations in the experiment section. The paper looks sound to me.

Presentation: The context of RAT is not sufficiently discussed. As a updated version of a recent work, most of the readers are not familiar with RAT. The method section should introduce the overall algorithm from scratch.

Significance: The paper gives a training-lightweight method to accelerate the long sequence inference. Ideally, the architecture research focuses on the lossless acceleration. while RAT+ achieves a promising trade-off between performance and throughput, it is still lossy method. It is questionable wheter RAT+ will be a general architecture. Given the lossy context, there are lots of related methods, including sparsification and KV compression methods.

Originality: Most of the methods are inherited from RAT. The paper's main contribution is giving a pre-training model adaption setting.

---

> ### Author Rebuttal · Authors · 2026-03-30
>
> We thank the reviewer for the constructive feedback and helpful suggestions. Here are the responses.
>
> 1.Significance and Q1: Thank you for the question. We would first like to clarify that we do not claim RAT+ achieves lossless acceleration, but is extending the set of inference-time sparsity patterns by enabling dilated attention. Dilated attention reduces FLOPs for both prefilling and decoding, saves KV cache storage, and keeps long-range dependency, which cannot be achieved by many other sparse variants simultaneously. We also train a 7B RAT+ to demonstrate its scalability and found an even smaller loss gap between the dilation and dense variants (see our first response to Reviewer AFwh). Moreover, another point we highlighted is that RAT+ also benefits the top-k block pattern.
>   * To our knowledge, for inference-time sparsity, only sliding window attention reduces FLOPs in both prefilling and decoding, as well as KV cache storage. Thus, we mainly compare with it (StreamingLLM) in Table 5.
>   * Top-k block attention (e.g., Quest and MoBA, which do not reduce KV cache storage) evaluated on 8 Needle-In-Haystack tasks in the Ruler benchmark:
>      - **In Figure 2, Table 11-12, we showed that applying them to RAT+ significantly outperforms applying to standard attention**, for example, 14% accuracy boost on NIAH-MK-3 for block size 16 and K=64, 89% increase with K=16. Our ablation study showed the benefit comes from the recurrence.
>      - **We also evaluated dilated attention on the same tasks and thus provided comparisons in Appendix C (Tables 10–12).** For example, RAT+ (D=4) achieves 88.8 accuracy on NIAH-MK-3. The accuracy performance of top-k block attention is highly sensitive to the block size, which also affects FLOPs overhead (e.g., block size 16 incurs T/16 additional FLOPs for block scoring) and contiguous memory access. Under the same KV eviction per token, its best performance reaches 99.2 with block size 16, but drops to 51.6 with block size 64.
>      - **This pattern can be orthogonal to our dilated pattern.** We show below that ours can help with its hard 64 block size setting by using together.
> ||||
> |-|-|-|
> |Method (D in top-k pattern means block size)|NIAH-MK-2|NIAH-MK-3|
> |Attention (D=64, K=16)|63.2|15.2|
> |RAT+ (D=64, K=16)|93.8|51.6|
> |RAT+ (D=64 \| D=64, K=16)|**99.2**|**97.0**|
>   * We add comparisons with SnapKV [1] in the rebuttal, which reduces KV cache storage as well but only accelerates decoding. We found they perform badly, which is consistent with results in a hybrid design paper [2].
> ||||||||||
> |-|-|-|-|-|-|-|-|-|
> |Method (same KV cache reduction below) |NIAH-S-1|NIAH-S-2|NIAH-S-3|NIAH-MK-1|NIAH-MK-2|NIAH-MK-3|NIAH–MQ|NIAH-MV|
> |RAT+ (D=4)|100|100|97.8|99.8|97.4|88.8|99.75|99.5|
> |StreamLLM (W=1024)|24.8|32.4|33.6|37.2|26.8|24.4|32.4|32.5|
> |SnapKV|57.6|64.2|13.2|56.4|41.2|7.2|46.3|50.85|
>
>    We will also update the revision accordingly to make our points clearer.
>
>    [1]. SnapKV: LLM Knows What You are Looking for  Before Generation
>
>    [2]. Ada-KV: Optimizing KV Cache Eviction by Adaptive Budget Allocation for Efficient LLM Inference
>
> 2.Presentation: We already have an extensive discussion of the RAT in the paper such as in Preliminary and Related work, but we agree to further clarify the context of RAT. In the revision, we will expand it and highlight differences between RAT and RAT+ more clearly (specifically the overlapped chunk size and active recurrence learning).
>
> 3.Originality: We think the originality of RAT+ is the new paradigm for efficient architecture, and the simplicity and effectiveness of the two proposed techniques. While many architecture papers focus on training efficient architecture from scratch, we propose to **design a better dense model that supports a broader set of inference sparsity patterns with high performance, which, to the best of our knowledge, has not been explored before**. Also, though techniques in RAT+ are simple and may look trivial in hindsight, **identifying the underlying challenges and developing effective solutions is non-trivial**. Works like LoRA and ResNet were also once considered unoriginal and simple, but have been very impactful in the community.
>
> In detail, in section 3.1, we demonstrate the importance of the recurrence from both training and inference aspects to build strong motivation. In 3.2, we notice that the different output distributions of recurrence at different short lengths bring difficulty for subsequent attention, and propose the overlapped chunk size. We then observe a lazy learning phenomenon and propose active recurrence learning to ensure sufficient optimization of both modules. Table 3 and 7 show the effectiveness of each technique. None of these points is covered by RAT, and as an efficient architecture, it can not easily adapt to different dilations.
>
> Thus, considering the new efficiency paradigm and the simple yet useful techniques, we hope we address the reviewer’s concern.

---

> > ### Author Rebuttal · Reviewer_j5q7 · 2026-04-02
> >
> > Thanks for your response. If your context is on the efficiency improvement of prefill, inference, and KV cache, there is actually few related works achieving these goals at the same time. However, since the method is a lossy acceleration, there are many lossless methods acheving maybe one or two goals. It's really difficult to give a fair comparison between these two variants.
> >
> > I personally still have doubt about this motivation, but as an academic research, the paper presentation looks fine. I will keep the score as a weak accept.

---

> > > ### Author Response · Authors · 2026-04-02
> > >
> > > Thank you for your quick reply and for maintaining a positive view of our paper. **To ensure nothing was overlooked in our rebuttal, we would like to clarify that we have provided comparisons with top-k block attention (e.g., Quest and MoBA with prefilling and decoding FLOPs reduction) and StreamingLLM (prefilling, decoding, KV cache) in the paper. We also added comparisons with SnapKV (decoding, KV cache) in the rebuttal. In addition, we highlight in the paper that using top-k block attention in the dense RAT+ achieves better results than in standard attention, with ablations demonstrating that recurrence also benefits other sparsity patterns.** Please refer to our first response for more details.
> > >
> > > We further provided results of **pretraining the 7B model to Reviewer AFwh, showing almost no performance degradation with up to an 8× computation reduction across multiple benchmarks. Note that the eight NIAH tasks from the Ruler benchmark we evaluated are very challenging, and many methods that might be lossless on simpler tasks show poor performance here** as can be seen in a hybrid design paper [2] and our comparisons above. In practice, there is generally a trade-off: methods with no accuracy degradation tend to achieve smaller speed-ups, while larger speed-ups often come at the cost of increased loss. RAT+ supports a range of dilation configurations, enabling favorable trade-offs between speed-up and accuracy.
> > >
> > > Finally, we would like to highlight the key perspective of our work. Compared to efficient architectures trained from scratch, we propose a new way of exploring efficiency by designing a more capable dense model, which enables more flexible inference-time behavior. Compared to standard attention, RAT+ supports different dilated attention patterns and their hybrid variants (with a short 1B-token resolution adaptation), as well as better results for the top-k block sparsity pattern. We show that dilated attention can be an important inference-time sparsity pattern, with strong efficiency, accuracy (as demonstrated in the paper and rebuttal), and scalability (e.g., results on the 2.6B and 7B model). Our method is built upon detailed analyses, from the motivation for incorporating recurrence to the challenges of enabling flexible inference. We also hope our work can provide useful insights for future efficiency work.
> > >
> > > We thank the reviewer for the time and engagement in improving our work. The results and analyses added in the rebuttal will be included in the revision.

---

### Official Review · Reviewer_AFwh · 2026-03-09

**Soundness:** 3
**Presentation:** 3
**Significance:** 3
**Originality:** 3
**Overall Recommendation:** 5
**Confidence:** 3

**Summary:**

This paper introduces a dense-pretraining architecture that augments standard attention with full-sequence recurrence (L=T) and active recurrence learning (ARL). Key insight being that dilated attn requires an explicit mechanism to construct a complete receptive field vanilla attn, and that a simple forget-rate recurrence over K/V can providfe it. RAT+ outperforms attn under top-k block sparsification on certain tasks.

**Compliance With Llm Reviewing Policy:**

Affirmed.

**Key Questions For Authors:**

Q1 - What's the parameter overhead of the recurrence module (gate + rec. projection) relative to base attn block? If its available in appendix, please point me to it in case I missed it.

Q2 - Is there evidence that the recurrence learns a smooth capacity across intermediate values, or does the performance plateau/cliff at specific D-boundaries? Fig 6 seems to have a version of this analysis but only for L*.

Q3 - For the top-k block result, you're saying "recurrence encourages token reps to better reflect the content of their block". This is interesting, but it seems a bit speculative? Have you measured this directly e.g. via block level rep. similarity or information content analysis? Consider limiting this claim if you have not.

**Limitations:**

Yes

**Strengths And Weaknesses:**

## Strengths

S1 - Paper provides a clearly identified failure mode with clean experimental isolation. Motivational analysis is well-structured and convincing.

S2 - Comprehensive evals, authors go beyond FLOPsto report actual wall-clock speed-ups and full model throughput. I think this systems level validation is well grounded.

S3 - Finding that RAT+ reps are more amenable to block-level scoring (fig 2) is interesting and suggests that recurrence may have broader benefits for importance based sparsification beyond dilation. I think this opens up an interesting direction in and of itself.

## Weaknesses

W1 - Limited scale/data for the claims being made, primary experiments are on models of (relatively) small scale. The central thesis of train dense, infer sparse is a systems level paradigm in my view, but at this scale it remains unclear whether the recurrence overhead and accuracy/efficiency tradeoff holds at scales where efficiency actually matters.

W2 - Commonsense benchmarks are fairly short-context, and that may mask the real challenge. I see the LongBench results but they're limited as well in my view.

W3 - Comparison to Mamba2 and GatedDeltaNet is reasonable, but these are fixed efficiency models (i.e. not train dense infer sparse). I think a better comparison would be against a Jamba-style architecture or other systems that interleave attention and recurrence/SSM layers, which may offer flexible efficiency knobs.

---

> ### Author Rebuttal · Authors · 2026-03-31
>
> We thank the reviewer for the positive feedback and insightful suggestions that help improve our work. Here are our responses:
>
> 1.W1: This is an important question. The efficiency of the 7B model (H=4096) is reported in Fig. 3–4 and Tab. 16–17. Since this is a simple recurrence and dilation reduces both FLOPs and KV cache, we achieve good speed-up in long-context prefilling, decoding, and throughput. For accuracy, we train a 7B model on 100B tokens and find more promising results.
>   *  The table below further strengthens our analysis that as model size increases, the gap between dense and dilated variants decreases, demonstrating the scalability of RAT+.
> ||||
> |-|-|-|
> ||Loss gap between D=1 and D=64|
> |1.5B (100BT)|0.035|
> |2.6B (100BT)|0.029|
> |2.6B (200BT)|0.028|
> |7.6B (100BT)|0.021|
>   * This is also confirmed in downstream evaluations. D=16,W=256 (8× reduction in temporal-mixing FLOPs and KV cache) even outperforms dense attention on some tasks. On NIAH-MK-2, D=32 achieves 78% accuracy at 1.5B but 94.6% at 7.6B. For all cases, scaling to 7.6B improves most tasks, except RBP in LongBench, likely due to few code pieces in the trainset.
> |||||||
> |-|-|-|-|-|-|
> |RAT+(7.6B)|Avg. CommonSense|Avg. LongBench|Avg. NIAH-S|Avg. NIAH-MK|Avg. (MQ,MV)|
> |D=1|64.4|20.8|100.0|99.8|100.0|
> |D=2|64.4|20.9|99.9|98.5|99.9|
> |D=4|64.2|21.2|99.9|97.1|99.8|
> |D=8|64.1|20.1|99.5|95.5|99.8|
> |D=16|63.6|20.1|98.8|90.5|99.4|
> |D=32|63.3|20.6|97.6|85.1|99.2|
> |D=64|63.1|20.0|97.6|79.1|98.6|
> |D=16, W=256|64.7|21.1|99.6|93.9|99.6|
>   * We also list existing 7B models here for reference for dense RAT+. Note that training on different datasets gives different performance.
> ||||||||
> |-|-|-|-|-|-|-|
> |Model|ARC-C|ARC-E|Hella.|LMB.|PIQA|Wino.|
> |Griffin(300BT)|47.9|75.4|78.6|-|81.0|72.6|
> |Pythia(300BT)|35.2|67.3|63.9|67.2|76.2|61.3|
> |Pythia(100BT)|29.3|62.3|56.4|61.2|73.6|56.8|
> |RAT+(100BT)|47.9|77.6|68.2|56.6|77.0|61.7|
>
> 2.W2: We showed dilated attention results on 8 Needle-in-Haystack tasks (4K context) in the Ruler benchmark (Table 10, Appendix C). Even with dilation size 64, we reach near-100% accuracy on most tasks, though some remain challenging. But as shown above, scaling model size further improves performance.
>
> 3.W3: We checked the Jamba paper and found that it still trains efficient architectures from scratch by experimenting with different numbers of Mamba and Transformer layers. So they are not necessarily “train dense infer sparse”.
>
> 4.Q1: The only difference from the base attn block is a linear projection with parameters H^2 (model dimension H) used for the forget-gate-like recurrence, as described in Lines 299–305; parameter saving is left for future work. Attention output gating is applied to both the attention and RAT+, as it has recently proven very effective (e.g., SSM, Qwen3-Next). The efficiency study is also conducted consistently (Figure 4, Table 14-17), with D† indicating no such projection.
>
> 5.Q2: Table 3 and Figure 6 show smooth changes in loss/PPL across dilation values (D), and we have not seen a performance cliff. Figure 6 plots loss vs. D for different L*: overlapped chunk sizes cover the range (D ≤ L*), and dilated attention gives smooth loss change. Beyond this range (D > L*), the loss remains smooth (gray points) because of the recurrence’s length generalization, although at higher values than using larger L* directly. Results for D=128 (L*=64) in Tab. 4 and 5 further support it.
>
> 6.Q3: Thank you for pointing this out. We agree that the original statement may read speculative. To provide more evidence, we analyze block scoring performance—based on block representatives and serving as the first step of top-k block patterns—as a proxy for representation quality. If recurrence improves representations, it should yield more accurate block selection. In the revision, we will update the sentence to: “Studies in Figure 2 and Table 11, 12 with ablations show that the performance gain comes from recurrence. We hypothesize that improved block scoring is one key factor and provide head-level analysis.”
>
> We evaluate this on retrieval tasks requiring the model to locate a long digit/UUID among similar key-value pairs. We measure each head’s hit rate, defined as whether the ground-truth token falls within the selected top-k blocks. Across 500 examples, a subset of heads consistently achieves high hit rates, aligning with prior findings [1] that only a few heads are critical for retrieval tasks. Compared to attention, top heads in RAT+ show much higher hit rates, indicating improved block scoring. A more complete understanding of the underlying mechanisms is left to future work.
> ||||||||
> |-|-|-|-|-|-|-|
> ||Acc|Top-1 head hit rate|Top-2|Top-3|Top-4|Top-5|
> |NIAH-MK-2 (D=64,K=16)|||||||
> |Attention|57.4|89.2|87.0|86.8|86.0|85.2|
> |RAT+|93.8|97.0|93.8|93.2|93.2|90.8|
> |NIAH-MK-3 (D=16,K=16)|||||||
> |Attention|5.4|75.8|75.8|75.0|74.0|73.8|
> |RAT+|94.8|99.8|99.6|99.6|99.6|97.6|
>
> [1].Retrieval Head Mechanistically Explains Long-Context Factuality.

---

> > ### Author Rebuttal · Reviewer_AFwh · 2026-04-03
> >
> > I acknowledge the author's response and intend to maintain my current score.

---

> > > ### Author Response · Authors · 2026-04-04
> > >
> > > We thank the reviewer for the positive assessment and for acknowledging that the concerns have been addressed. We are very grateful for the reviewer’s feedback, especially the suggestion to train at the 7B model scale, which further strengthens our paper. We also appreciate the reviewer’s thought that our findings about recurrence and other sparsity patterns in the paper may open up an interesting direction, which has inspired us to deeply investigate it in future research.
> > >
> > > Finally, we would like to highlight the key perspective of our work. Compared to efficient architectures trained from scratch, we propose a new way of exploring efficiency by designing a more capable dense model, which enables more flexible inference-time behavior. Compared to standard attention, RAT+ supports different dilated attention patterns and their hybrid variants (with a short 1B-token resolution adaptation), as well as better results for the top-k block sparsity pattern. We show that dilated attention can be an important inference-time sparsity pattern, with strong efficiency, accuracy (as demonstrated in the paper and rebuttal), and scalability (e.g., results on the 2.6B and 7B model). Our method is built upon detailed analyses, from the motivation for incorporating recurrence to the challenges of enabling flexible inference. We also hope our work can provide useful insights for future efficiency work.

---

### Official Review · Reviewer_ePTi · 2026-03-19

**Soundness:** 2
**Presentation:** 2
**Significance:** 2
**Originality:** 2
**Overall Recommendation:** 4
**Confidence:** 3

**Summary:**

This paper studies the challenge of applying dilated attention in the context of post-training.
Standard methods typically require either training an efficient architecture from scratch or involve additional training after conversion, which leads to substantial overhead.
To address this, the authors proposed RAT+, a dense pretraining architecture built on top of RAT that can be flexibly switched to dilated attention at inference time.
To enable this, RAT+ uses full-sequence recurrence and active recurrence learning (jointly training with D=1 and D=64).
Experiments at 1.5B and 2.6B scales evaluate various sparse inference patterns on commonsense reasoning and long-context tasks, showing that a single RAT+ model can adapt to different dilation settings with modest acc loss and significant throughput gains.

**Compliance With Llm Reviewing Policy:**

Affirmed.

**Final Justification:**

Most of my concerns have been addressed during rebuttal period. So I'm willing to raise my score.
However, I want to note that the "train once, deploy anywhere" framing is somewhat misleading, since each dilation setting still requires a separate 1B-token adaptation. This is not addressed in the rebuttal.

**Key Questions For Authors:**

1. Now each dilation setting requires a separate 1B-token adaptation  How sensitive are the downstream results to the adaptation budget? Would a single multi-dilation adaptation (e.g. training with mixed D values simultaneously) work, to truly achieve "train once, deploy anywhere"?
2. Recent training-free sparse attention methods achieve comparable or better speedups without requiring recurrence or per-dilation adaptation. And they can also select different settings during inference. What advantages does the dilated attention + recurrence approach offer over these alternatives? A direct comparison would significantly strengthen the paper.
3. The active recurrence learning uses D=64 as the sparse training target. Could the author explain why this specific value was used? Is the method sensitive to this choice?

**Limitations:**

The authors partially discussed limitations in the future work section.

**Strengths And Weaknesses:**

Strengths:
1. The motivation of the paper is straightforward, and the method is quite intuitive. This concept of being able to train once and then deploy across various settings is quite interesting.
2. Each design choice is cleanly isolated by ablation studies.
3. The efficiency results are great. On an end-to-end level, setting D = 64 achieves a 60x end-to-end throughput speedup on 1.5B model.

Weaknesses:
1. The main objective of this paper is to make dilated attention work from a dense model, but never justifies why dilated attention deserves this investment. Recent methods like MoBA, NSA, DSA and other training-free inference speedup approaches are not compared. To my knowledge, there are currently no production-level LLMs utilizing this type of dilated attention. Please provide more direct comparisons or explanations as to why this type of attention is better than context-aware sparse attention.
2. Main experiments has systematic gaps. For example, in the commonsense results in Table 4, the authors mentioned that the lengths on these tests are all under 300, where large dilation is essentially irrelevant. (And the parameter count for RAT+ model and Mamba2 seems to be different.) However, on the LongBench results in Table 5 (where it should actually matter), there are only results for RAT+ and no results for other baselines. I would like to see baseline performances on LongBench.
3. The contribution is incremental over RAT. The individual components are all known. The experiments the author conducted in the third section are quite trivial and don't seem to provide any meaningful insight.

---

> ### Author Rebuttal · Authors · 2026-03-30
>
> We thank the reviewer for insightful feedback and suggestions in making our work better. Here are our responses:
>
> 1.W1 and Q2: This is an important question. As Reviewer j5q7 raised a similar concern, please see our first response there for details. Here we provide a brief summary and will make the revision clearer.
>
> Dilated attention reduces FLOPs in both prefilling and decoding, reduces KV cache storage, and keeps long-range dependencies—properties that many sparse variants cannot achieve simultaneously. We train a 7B model and found an even smaller loss gap between dense and dilated variants (refer to first response to Reviewer AFwh) to further demonstrate its scalability. Moreover, RAT+ benefits the top-k block pattern as well.
>   * Sliding window attention is the main existing method with the three efficiency benefits. We compared it (StreamingLLM) in Tab. 5.
>   * Top-k block attention (e.g., Quest, MoBA; No KV cache storage reduction):
>     - Applying them to RAT+ significantly outperforms applying to standard attention (Fig. 2, Tab. 11, 12).
>     - Appendix C (Tab. 10–12) includes comparisons with dilated attention on the same tasks. MoBA is originally used only in prefilling; we apply it in both stages for fairness.
>     - NSA and DSA are not training-free: DSA requires continued pretraining (~1TB tokens), while NSA trains from scratch with dilated, local, and top-k patterns. We add results of supporting this pattern at inference by RAT+.
>   * We add results of SnapKV, which reduces KV cache but only accelerates decoding.
>
> Dilated attention has been widely studied (e.g., LongNet, Longformer, BigBird) and is also used in recent NSA trained from scratch. Together with the above results, we believe that it is important to have RAT+.
>
> 2.W2: Table 4 involves short contexts (<300 tokens) without truncation. In this regime, large dilation almost reduces to simple recurrence with vector-sized memory, reflecting performance at the extreme setting. Although we add one extra linear projection for recurrence, the matrix-sized memory with 256 slots used by Mamba2 exceeds even RAT+ (D=2). As a fixed architecture, we include it mainly as a reference to illustrate our high accuracy.
>
> Since RAT+(D=1) performs best in Table 4, we use it as the baseline in Tab. 5 to explore different patterns. We compare with Mamba2 below under the same KV cache budget. By compressing the full sequence, it performs worse than ours, which retains direct access to past tokens.
> ||||||||
> |-|-|-|-|-|-|-|
> ||NQA|Qasper|MF|HQA|2WQA|RBP|
> |RAT+(D=16)|14.72|14.54|25.99|14.68|17.4|23.68|
> |Mamba2|11.1|11.3|18.6|11.8|15.1|20.6|
>
> 3.W3: We think the originality of RAT+ is our new paradigm in exploring efficient architecture, and the simplicity and effectiveness (ablation studies in Tab. 3 and 7) of proposed two techniques. Compared to training efficient models from scratch, we design a better dense model that supports a broader set of inference-time sparsity patterns. Also, though our techniques may look simple in hindsight, identifying the underlying challenges and developing effective solutions is non-trivial. Please see our last response to Reviewer j5q7 for details. We hope this can address the concern.
>
> 4.Q1: In Fig. 7, we presented loss curves of the 1B-token resolution adaptation process and it is not sensitive. Most cases reach stable loss in the first 50 training steps, and 500M tokens are enough for all. We reserve 1B token just for safe results. In the motivation section and Tab. 2, we showed that this adaptation does not come from recurrence, but already exists when sparsifying attention to dilated patterns. Potential reasons were discussed in lines 145–151.
>
> For single multi-dilation adaptation with total token budget as 2B, we find that if only a few dilation settings are required (e.g., sampling one dilation per model in each forward pass, second row), the model can quickly learn them together. Since RAT+ also supports hybrids, we further consider a much larger space (e.g., sampling dilation per layer, third row) and find it is more challenging.
> |||||||||
> |-|-|-|-|-|-|-|-|
> |Val PPL|D=1|D=2|D=4|D=8|D=16|D=32|D=64|
> |Baseline|7.40|7.42|7.44|7.48|7.52|7.58|7.66|
> |Sample dilation per model|7.40|7.42|7.44|7.48|7.52|7.58|7.67|
> |Sample dilation per layer|7.49|7.48|7.49|7.52|7.59|7.70|7.86|
>
> 5.Q3: We discussed this in Fig. 6 and the last paragraph in 5.3. Other choices also work. For the sparse case trained with a smaller dilation like 16, the model benefits from slightly more training FLOPs and better recurrence capability for handling shorter contexts. However, it supports the inference dilation up to 16, with larger dilations enabled by the recurrence's length generalization. Here, we adopt 64, which is generally considered easy to handle by recurrence while allowing more dilation choices. Larger models (e.g., 2.6B, 7.6B) support this better. Tab. 3 also shows that its dense performance matches pure attention models trained with the same FLOPs.

---

> > ### Author Rebuttal · Reviewer_ePTi · 2026-04-04
> >
> > Thank you for the rebuttal. I am willing to raise my score to 4.
> >
> > However, I want to note that the "train once, deploy anywhere" framing is somewhat misleading, since each dilation setting still requires a separate 1B-token adaptation. This is not addressed in the rebuttal. I would suggest the authors either tone down this claim or show that a single mixed-D adaptation can cover multiple dilation settings.

---

> > > ### Author Response · Authors · 2026-04-04
> > >
> > > We thank the reviewer for the follow-up and for raising the score to 4.
> > >
> > > Regarding Q1 on the multiple-dilation setting, we notice that the phrase “sample dilation per model” in our rebuttal may be ambiguous. **What we intended to convey is that, during the adaptation phase, dilation is sampled for the full model in each forward pass. Under this setting, we train with the 7 dilation choices jointly during adaptation (second row), and we find that this performs well, achieving the same results as the baseline (first row), where each choice is adapted separately. Therefore, this shows a single mixed-D adaptation can cover multiple dilation settings**. The more challenging setting shown in the third row instead attempts to train $7^{24}$ possible dilation configurations jointly in the adaptation by sampling dilation for each layer, in order to cover an extremely large space ($\sim10^{20}$) of hybrid choices. This scale is very difficult for the model to memorize each configuration through simultaneous adaptation, since it only has $\sim10^{5}$ forward passes ($\ll 10^{20}$), making it impossible to see each choice even once.
> > >
> > > For presentation, we fully agree that this is an important point. We have emphasized the short 1B-token adaptation in the paper with nine times appeared in the main text, but we agree to highlight it more when mentioning "train dense, infer sparse". Also in the revision, we plan to move the adaptation loss curves in Figure 7, together with the table shown in the rebuttal, into the main text under a paragraph titled “Adaptation Analysis" in section 5.3. There, we will highlight three points:
> > > (1) the adaptation phase is indeed very short, and 1B tokens are already sufficient;
> > > (2) **a single mixed-dilation adaptation works well when covering several cases**, but becomes difficult when aiming to cover all possible cases because of the large space of hybrid choices, and the model fails to see each even once during adaptation; and
> > > (3) although this adaptation phase arises from the attention mechanism itself, fully addressing it remains an important direction for future work.
> > >
> > > We hope this can address the reviewer's concern.  Finally, we would like to highlight the key perspective of our work. Compared to efficient architectures trained from scratch, we propose a new way of exploring efficiency by designing a more capable dense model, which enables more flexible inference-time behavior. Compared to standard attention, RAT+ supports different dilated attention patterns and their hybrid variants (with a short 1B-token resolution adaptation before inference), as well as better results for the top-k block pattern.  We show that dilated attention can be an important inference-time sparsity pattern, with strong efficiency, accuracy (as demonstrated in the paper and rebuttal), and scalability (e.g., results on the 2.6B and 7B model). Our method is built upon detailed analyses, from the motivation for incorporating recurrence to the challenges of enabling flexible inference. We also hope our work can provide useful insights for future efficiency work.
> > >
> > > We sincerely thank the reviewer for the time, thoughtful comments, and discussions, which have helped improve our work. The results and analyses in the rebuttal will be included in the revision, and the presentation will be modified accordingly.

---

### Decision · Program_Chairs · 2026-04-30

**Decision:**

Accept (regular)

**Comment:**

This paper tackles a problem of previous dilated attention which requires either training an efficient architecture from scratch or spending extensive additional training after converting to dilated mode. To address this, the paper pretrains a dense architecture RAT+ (built on top of RAT) which update keys and values by explicitly modeling recurrence (like a gated RNN over sequences of key and value). This architecture (verified on 1.5B and 2.6B scales) is shown to easily adapted to various sparse inference patternswith only 1B token additional fine-tuning, demonstrating up to 60 times speedup over dense counterpart. The reviewers unanimously gave positive scores.